# CONDITIONAL GENERATIVE MODELS ARE SUFFICIENT TO SAMPLE FROM ANY CAUSAL EFFECT ESTIMAND

## ABSTRACT

Causal inference from observational data has recently found applications in machine learning applications. While there exist sound and complete algorithms to compute causal effects, these algorithms require explicit access to conditional likelihoods over the observational distribution. In the high dimensional regime, conditional likelihoods are difficult to estimate. To alleviate this issue, researchers have approached the causal effect estimation problem by simulating causal relations with neural models. However, none of these existing approaches can be applied to generic scenarios such as causal graphs having latent confounders and obtaining conditional interventional samples. In this paper, we show that any identifiable causal effect given an arbitrary causal graph containing latent confounders can be computed through push-forward computations using trained conditional generative models. Based on this observation, we devise a diffusion-based approach to sample from any such interventional or conditional interventional distribution. To showcase our algorithm's performance, we conduct experiments on a semi-synthetic Colored MNIST dataset having both the intervention ($X$) and the target variable ($Y$) as images and present interventional image samples from $P(Y|do(X))$. We also perform a case study on a real-world COVIDx chest X-ray image dataset to demonstrate our algorithm's utility.

## 1 INTRODUCTION

Causal inference has been a central problem in many sciences and is also recently understood to be critical for developing more robust and reliable machine learning solutions (Xin et al., 2022; Zhang et al., 2020; Subbaswamy et al., 2021). Although randomized controlled trials are known as the gold standard for estimating causal effects, modern formalisms such as the structural causal model (SCM) framework (Pearl, 2009) enable a data-driven approach to this problem: Given the qualitative causal relations, summarized by a *causal graph*, certain causal queries can be uniquely identified from the observational distribution. Today, we have a complete understanding of which causal queries can be uniquely identified, and which require further assumptions or experimental data, given this structure (Tian, 2002; Shpitser & Pearl, 2008; Bareinboim & Pearl, 2012).

For certain causal structures, sample-efficient ways to estimate (average) causal effect exist through, for example using propensity scores, or backdoor adjustments (Rosenbaum & Rubin, 1983; Pearl, 1993; Maathuis & Colombo, 2015). However, the most general solutions assume that we have access to the joint observational probability distribution of data. For example, the sound and complete causal effect identification algorithm of Shpitser & Pearl (2008) writes interventional distributions as functions of the observational distribution. These functions can seemingly get arbitrarily complicated and in general there is no easy way to directly estimate them from data.

This creates an important gap between causal inference and modern ML datasets: We typically observe high-dimensional variables, such as X-ray images of a patient, that need to be involved in causal effect computations. However, explicit likelihood-based models are impractical for such high-dimensional data. Instead, deep generative models have shown tremendous practical success in correctly *sampling* from such high-dimensional variables (Brock et al., 2018; Karras et al., 2019;

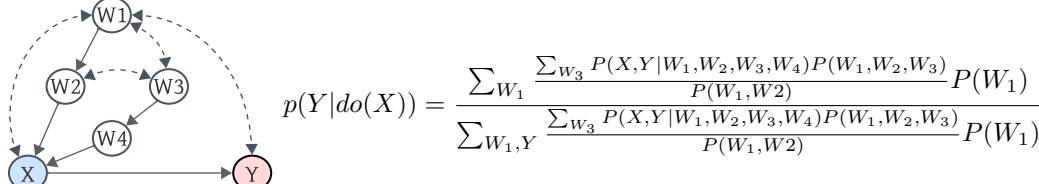

$$p(Y|do(X)) = \frac{\sum_{W_1} \frac{\sum_{W_3} P(X,Y|W_1,W_2,W_3,W_4)P(W_1,W_2,W_3)}{P(W_1,W2)}P(W_1)}{\sum_{W_1,Y} \frac{\sum_{W_3} P(X,Y|W_1,W_2,W_3,W_4)P(W_1,W_2,W_3)}{P(W_1,W2)}P(W_1)}$$

Figure 1: Double napkin graph and the corresponding causal effect estimand for $P(Y|do(X))$.

Croitoru et al., 2023). We are interested in the question: Can we leverage the representative power of deep conditional generative models for causal inference?

As an example consider the causal graph in Figure 1. We are interested in the distribution of variable $Y$ after a hypothetical intervention on $X$. We use the do-operator of Pearl to represent the intervention, i.e., we are interested in $p(Y|do(X))$. The ID algorithm (Shpitser & Pearl, 2008) identifies this distribution in terms of the observational distribution as given on the right. This complicated expression can be evaluated if one has explicit access to the joint distribution $P$. However, it is not clear if one can sample from this probability distribution without an explicit likelihood-based model.

In this paper, we show that this is possible. Specifically, we show that any identifiable interventional distribution can be sampled from by only training conditional generative models on the observational data. This enables us to leverage the state-of-the-art deep generative models, such as diffusion models, and even sample from $p(Y|do(X))$ when both $X$ and $Y$ are high dimensional. To the best of our knowledge, this is the first result that shows that conditional generative models are sufficient to sample from any identifiable interventional distribution. Our contributions are as follows:

- We propose a training algorithm that learns a collection of conditional generative models using observational data and can sample from any identifiable interventional distribution.

- We show that our algorithm is sound and complete, establishing that conditional generative models are sufficient to sample from any identifiable interventional query. The guarantees extend to sampling from any identifiable conditional interventional distribution.

- We use diffusion models to demonstrate the performance of our algorithm on a synthetic image dataset, as well as a real-world COVIDx chest X-ray dataset.

## 2 RELATED WORK

Over time, extensive literature has developed on the causal effect estimation problem. Given access to the causal graph and the probability distribution, Pearl proposed the do-calculus (Pearl, 1995) rules as a general solution for any identifiable causal effect. Shpitser & Pearl (2008) propose the popular sound and complete identification algorithm to express the causal effect of an arbitrary set of variables on other variables in terms of observational distributions or suggest a graphical criterion for non-identifiability. Bareinboim & Pearl (2012); Jaber et al. (2018); Lee et al. (2020) have proposed variants of the causal estimation problem based on different restrictions on the causal graph and the access to probability distributions. However, most of these studies have relied on explicit access to the observational probability distributions which limits their applicability to be employed in causal inference problems with high dimensional data.

The use of deep neural networks for performing causal inference has been recently suggested by many researchers. Shalit et al. (2017); Louizos et al. (2017) proposed neural network-based approaches to estimate the causal effect. Sanchez & Tsaftaris (2022) employs energy-based generative models such as DDPMs (Ho et al., 2020) to generative high dimensional interventional samples. However, solutions of these methods do not generalize to arbitrary structures.

The sampling-based approaches using deep generative models are limited in the literature. Kocaoglu et al. (2018) trains a collection of conditional generative models based on the causal graph and uses adversarial training. Pawlowski et al. (2020) employed a conditional normalizing flow-based approach to offer high dimensional interventional sampling as part of their solution. Chao et al. (2023) performs interventional sampling for arbitrary causal graphs employing diffusion-based causal mod-

els with classifier-free guidance (Ho & Salimans, 2022). However, all of these methods depend on the strong assumption that the system has no unobserved confounders.

Xia et al. (2021; 2023) extends this idea to not only sample from interventional distribution but also test the identifiability of the interventional distribution. However, their approach does not handle high-dimensional variables. Perhaps more importantly, all these works train a forward model based on the causal graph structure. Conditional sampling then becomes tricky since it is not clear how to update the posterior of upstream variables using feedforward operations. For example, Kocaoglu et al. (2018) resorts to rejection sampling, which is slow and impractical for high-dimensional data.

Perhaps the most conceptually related work is Jung et al. (2020), where the authors identified an algorithm that can convert the expression returned by the ID algorithm into a form where it can be computed through a re-weighting function, similar to propensity score-based methods, to allow sample-efficient estimation. However, computing these reweighting functions from data is still highly nontrivial with high-dimensional variables in the system.

## 3 BACKGROUND

We first introduce the structural causal models (SCMs) and how they can model interventions.

**Definition 3.1** (Structural causal model, (SCM) (Pearl, 1980)). An SCM is a tuple $\mathcal{M} = (G = (\mathcal{V}, \mathcal{E}), \mathcal{N}, \mathcal{U}, \mathcal{F}, P(.))$. $\mathcal{V} = \{V_1, V_2, ..., V_n\}$ is a set of observed variables in the system. $\mathcal{N}$ is a set of independent exogenous random variables where $N_i \in \mathcal{N}$ affects $V_i$ and $\mathcal{U}$ is a set of unobserved confounders each affecting any two observed variables. This refers to the semi-Markovian causal model. A set of deterministic functions $\mathcal{F} = \{f_{V_1}, f_{V_2}, ..., f_{V_n}\}$ determines the value of each variable $V_i$ from other observed and unobserved variables as $V_i = f_i(Pa_i, N_i, U_{S_i})$, where $Pa_i \subset \mathcal{V}$ (parents), $N_i \in \mathcal{N}$ (randomness) and $U_{S_i} \subset \mathcal{U}$ (common confounders) for some $S_i$. $\mathcal{P}(.)$ is a product probability distribution over $\mathcal{N}$ and $\mathcal{U}$ and projects a joint distribution $\mathcal{P}_{\mathcal{V}}$ over the set of actions $\mathcal{V}$ representing their likelihood.

An SCM $\mathcal{M}$, induces an acyclic directed mixed graph (ADMG) $G = (\mathcal{V}, \mathcal{E})$ containing nodes for each variable $V_i \in \mathcal{V}$. For each $V_i = f_i(Pa_i, N_i, U_{S_i})$, $Pa_i \subset \mathcal{V}$, we add an edge $V_j \rightarrow V_i \in \mathcal{E}, \forall V_j \in Pa_i$. Thus, $Pa_i(V_i)$ becomes the parent nodes in $G$. $G$ has a bi-directed edge, $V_i \leftrightarrow V_j \in \mathcal{E}$ between $V_i$ and $V_j$ if and only if they share a latent confounder. If a path $V_i \rightarrow \ldots \rightarrow V_j$ exists, then $V_i$ is an ancestor of $V_j$, i.e., $V_i = An_G(V_j)$. An intervention $do(v_i)$ replaces the structural function $f_i$ with $V_i = v_i$ and in other structural functions where $V_i$ occurs. The distribution induced on the observed variables after such an intervention is represented as $\mathcal{P}_{v_i}(\mathcal{V})$. Graphically, it is represented by $G_{\overline{V_i}}$ where incoming edges to $V_i$ are removed.

**Definition 3.2** (c-component). Given an ADMG a maximal subset of nodes where any two nodes are connected by bidirected paths is called a c-component. $C(G)$ is the set of c-components of $G$.

Pearl (1995) identified do-calculus rules, which relate different interventional and observational distributions to one another. Namely, rules identify conditions on the causal graph that allow $i$) removing/adding conditioning variables in the probability distribution (rule-1), $ii$) replacing do-operator with conditioning (rule-2) and $iii$) removing/adding do-operations (rule-3). These form the basis for any identification algorithm, and were shown to be complete to identify any identifiable interventional distribution (Shpitser & Pearl, 2008; Huang & Valtorta, 2012). In other words, finitely many applications of these three rules is sufficient to convert an interventional distribution into a function of the observational distribution, if possible. Shpitser & Pearl (2008); Tian & Pearl (2002); Tian (2002); Huang & Valtorta (2012) provide systematic ways to apply them.

**Lemma 3.3** (c-component factorization (Tian & Pearl, 2002)). *Let $\mathcal{M}$ be an SCM that entails the causal graph $G$ and $P_x(y)$ be the interventional distribution for arbitrary variables $X$ and $Y$. Let $C(G \setminus X) = \{S_1, \ldots, S_n\}$. Then we have $P_x(y) = \sum_{v \setminus (y \cup x)} P_{v \setminus s_1}(s_1) P_{v \setminus s_2}(s_2) \ldots P_{v \setminus s_n}(s_n)$.*

Intuitively, $P_x(y)$ is factorized into *c-factors* $P_{v \setminus s_i}(s_i)$ for each c-component $S_i$ in $G \setminus X$.

**Diffusion models.** For the purposes of our algorithm, we aim to black-box the technical details of the generative models we learn. This is intentional: the field of conditional generative modeling is advancing rapidly and it is not at all clear that the current state-of-the-art frameworks will be used in the near future. For the purposes of our framework, we have the following simple requirement:

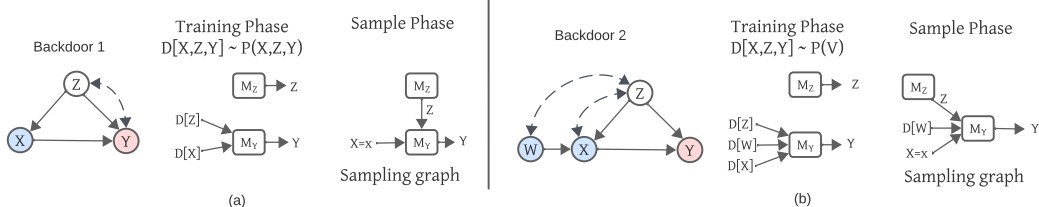

Figure 2: High dimensional interventional sampling: backdoor graph with $Y$ as an image

given samples from a joint distribution, we need to be able to learn a model that provides approximate samples from any conditional distribution. In practice, we will use classifier-free diffusion guidance (Ho & Salimans, 2021), but our framework is agnostic to the choice of generative model.

In this paper, we assume i) the causal model is semi-Markovian and ii) we have access to the ADMG, and iii) we assume that we can learn to sample high dimensional conditional distributions by training a classifier-free guidance diffusion model on samples from a joint distribution.

# 4    CONDITIONAL GENERATIVE MODELS FOR CAUSAL SAMPLING

Given a causal graph $G$, dataset $D \sim P(V)$, our objective is to generate samples from a causal query $P(Y|\mathrm{do}(X))$. Before we describe our algorithm formally, we express the challenges and outline our main ideas which lead to the general algorithm over examples.

## 4.1    CHALLENGES WITH CAUSAL INFERENCE VIA SAMPLING AND MOTIVATING IDEAS

As the first example, consider the backdoor graph in Figure 2(a). Suppose we have a dataset $\mathcal{D} \sim P(X, Y, Z)$. Consider the causal query $P_X(Y)$. An application of ID algorithm shows that $P_X(Y) = \sum_z P(Y|X, z)P(z)$. We now intuit how to sample from this distribution. Suppose, for example, that we could sample from $P_X(Y, Z)$; in other words suppose we had a mechanism that when provided a value for $x$ would provide samples $(y, z) \sim P_x(Y, Z)$. From this we can derive a way to sample from $P_X(Y)$, by sampling $(y, z) \sim P_x(Y, Z)$ and only keeping the $y$ variable: Dropping $Z$ from the joint sample is equivalent to sampling $Y$ from the distribution where $Z$ is marginalized out.

Now focusing on $P_X(Y, Z)$, ID gives $P_X(Y, Z) = P(Y|X, Z)P(Z)$. Sampling from this distribution seems more attainable; indeed if we could sample from both $P(Z)$ and $P(Y|X, Z)$, we could use the following procedure: first sample $z \sim P(Z)$ and then use this $z$ and the specified interventional $x$ to sample from $P(Y|x, z)$. Hence the only ingredients we need are sample access to $P(Z)$ and $P(Y|X, Z)$. From the joint dataset $\mathcal{D}$, we can train conditional generative models $M_Z$ and $M_Y$ to approximate sampling from these distributions. With these models in hand, they can be wired together in a sequential structure, which we visualize as a DAG, where each variable corresponds to a node and its (conditional) generative model. Sampling from this DAG can be performed by sampling from each node in a topological order, and passing sampled values to descendant nodes.

Next we consider the backdoor graph in 2(b), where we again want to sample from $P_X(Y)$. Applying the ID algorithm yields $P_X(Y) = \sum_z P(z)P(Y|X, W, z)$, which looks identical to the previous example, except for the added term $W$ in the conditional distribution. This raises the important question of where this $W$ comes from. Do-calculus ensures in this case that $P_X(Y) = P_{X,W}(Y)$; in other words, the causal effect of $W$ on $Y$ is irrelevant assuming we also intervene on $X$. Hence we can pick *any* value of $W$ to apply here. Then intuitively, sampling can proceed in a similar fashion to the previous example. First attain a sampling mechanism $M_Z$ which samples $z \sim P(z)$, and then attain a sampling mechanism $M_Y$ providing sample access $P(Y|X, W, Z)$. Sampling can then proceed by using $M_Z$ to get $z$, picking any arbitrary value $w$, and sampling $y$ from $M_Y$. Again this can be arranged in a DAG structure, where sampling can be done according to the topological order. Figure 2(b) demonstrates the graphical representation of this procedure.

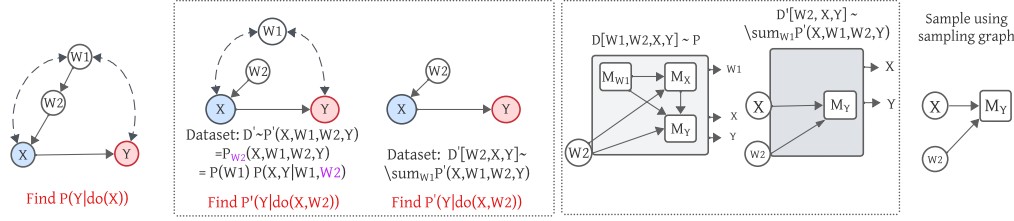

Figure 3: High dimensional interventional sampling from napkin graph

As our next example we consider the Napkin graph in Figure 3. For $P_X(Y)$, ID algorithm returns

$$P_X(Y) = \frac{\sum_{w_1} P(X, Y \mid W_1, W_2) P(W_1)}{\sum_{w1} P(X \mid W_1, W_2) P(W_1)}.$$

While this expression seems more complicated, it is helpful to note that, with another call to the ID algorithm, we can show that $P_{W_2}(X, Y) := \sum_{w_1} P(X, Y | W_2, W_1) P(W_1)$. Hence, if we define $P'(V) := P_{W_2}(V)$, sampling from $P_X(Y)$ is equivalent to sampling from $P'(X, Y)/P'(X) = P'(Y|X)$. In other words, if we could sample a dataset $\mathcal{D}' \sim P'(V)$, which implicitly depends on $W_2$, then we could sample from $P_X(Y)$ by learning a conditional model $M_Y$ using data from $P'(V)$ that emulates samples $y \sim P'(Y|X)$. We note that since $P'$ depends on $W_2$, $M_Y$ should be a function of $W_2$, so we can pass it as an argument to $M_Y$. This would be correct, only if we could generate a dataset $\mathcal{D}' \sim P'(V) = P_{W_2}(V)$. But sampling from an interventional distribution is precisely the problem we are seeking to solve: this suggests that a recursion to smaller subproblems must play a part in our algorithm, similar to the ID algorithm. On the other hand, we notice that $P_{W_2}(X, Y)$ has a similar structure to the backdoor graph, which we have already established we can sample from. Thus a sampling procedure for $P_X(Y)$ must involve several steps: generate a dataset $\mathcal{D}' \sim P_{W_2}(V)$, use $\mathcal{D}'$ to train $M_Y$, and then sample from $M_Y$.

We can summarize the insights from these examples as follows: for an identifiable query $P_X(Y)$ and access only to joint samples, we were able to generate a directed acyclic graph where each node corresponds to a variable and all but the interventional variables are imbued with a conditional generative model that only depends on its parents in the DAG. Sampling from the interventional query can be performed by sampling each node in this graph according to its topological ordering. The only ingredient we have required throughout this procedure is the ability to learn models to approximate samples from a conditional distribution given only joint sample access. We have been intentionally informal during these examples, but we formalize this procedure in the next section.

## 4.2 ID-DAG: A GENERATIVE MODEL-BASED ALGORITHM FOR INTERVENTIONAL SAMPLING

Now we are ready to present our algorithm in full generality. The key idea is that we will be given a causal query $P_X(Y)$, a dataset $\mathcal{D}$ from the joint distribution $P$, and a causal graph $G$. Our algorithm will follow the same recursive structure of the ID algorithm and return a DAG where each node corresponds to a variable and is associated with a conditional generative model that depends only on its parent nodes in that DAG. Sampling can then be performed by providing values for the intervened variables and calling each generative model according to the topological ordering of the DAG.

The first step of our algorithm is an identifiability check: we run ID on our provided causal query and if ID returns FAIL, then the query is not identifiable and we return FAIL. Otherwise, we proceed.

Our key contribution is the ID-DAG algorithm in Algorithm 1. We will spend the next several paragraphs outlining its operation. We note that, identical to the ID algorithm, ID-DAG proceeds through a series of 7 steps, several of which are recursive. The non-recursive steps are called base cases. In the recursive steps, we point out several key differences from the ID algorithm. Where ID passes distributions to its recursive calls, ID-DAG passes *datasets*, assumed to be joint samples from the corresponding distribution in the ID algorithm. Further, where ID returns arithmetic expressions of likelihood functions, ID-DAG returns *sampling networks*, i.e., a collection of conditional generative models that form a DAG structure. These need to be incorporated into a larger global sampling

---

**Algorithm 1** ID-DAG($\mathbf{Y}, \mathbf{X}, \mathcal{D}, G$)

---

1: **Input:** target $\mathbf{Y}$, intervention $\mathbf{X}$ , training data $\mathcal{D}$, causal graph $G$.
2: **Output:** A DAG of trained models.
3: Let $\pi$ be the topological order of nodes in $G$.
4: **if** $\mathbf{X} = \emptyset$ **then** {Step 1}
5:   $H = \emptyset$
6:   **for** each $V_i \in \mathbf{Y}$ in topological order **do**
7:     Let $M_{V_i}$ be a model trained on $\{V_i, V_\pi^{(i-1)}\} \sim \mathcal{D}$ such that $M_{V_i}(V_\pi^{(i-1)}) \sim P(V_i|V_\pi^{(i-1)})$
8:     Add node $(V_i, M_{V_i})$ to $H$
9:     Add edge $V_j \to V_i$ to $H$ for all $V_j \in V_\pi^{(i-1)}$
10:   **Return** $H$
11: **if** $\mathbf{V} \setminus An(\mathbf{Y})_G \neq \emptyset$ **then** {Step 2}
12:   **Return ID-DAG**$(\mathbf{Y}, \mathbf{X} \cap An(\mathbf{Y})_G, \mathcal{D}' = \mathcal{D}[An(\mathbf{Y})_G], G_{An(\mathbf{Y})})$
13: Let $\mathbf{W} = (\mathbf{V} \setminus \mathbf{X}) \setminus An(\mathbf{Y})_{G_{\overline{\mathbf{X}}}}$ {Step 3}
14: **if** $\mathbf{W} \neq \emptyset$ **then** {}
15:   **Return ID-DAG**$(\mathbf{Y}, \mathbf{X} = \mathbf{X} \cup \mathbf{W}, \mathcal{D}, G)$
16: **if** $C(G \setminus \mathbf{X}) = \{S_1, \ldots, S_k\}$ **then** {Step 4}
17:   **for** each $S_i \in C(G \setminus \mathbf{X}) = \{S_1, \ldots, S_k\}$ **do**
18:     $H_i =$**ID-DAG**$(S_i, \mathbf{X} = \mathbf{V} \setminus S_i, \mathcal{D}, G)$
19:   **Return** ConstructDAG$(\{H_i\}_{\forall i})$
20: **if** $C(G \setminus X) = \{S\}$ **then**
21:   **if** $C(G) = \{G\}$ **then** {Step 5}
22:     throw FAIL
23:   **if** $S \in C(G)$ **then** {Step 6}
24:     $H = \emptyset$
25:     **for** each $V_i \in \mathbf{X}$ **do**
26:       Add node $(V_i, \emptyset)$ to $H$
27:     **for** each $V_i \in S$ **do**
28:       Let $M_{V_i}$ be a model trained on $\{V_i, V_\pi^{(i-1)}\} \sim \mathcal{D}$ such that $M_{V_i}(V_\pi^{(i-1)}) \sim P(V_i|V_\pi^{(i-1)})$
29:       Add node $(V_i, M_{V_i})$ to $H$
30:       Add edge $V_j \to V_i$ to $H$ for all $V_j \in V_\pi^{(i-1)}$
31:     **Return** $H$.
32:   **if** $(\exists S')$ such that $S \subset S' \in C(G)$ **then** {Step 7}
33:     $\mathbf{X}_Z = \mathbf{X} \setminus S'$,    $H_{S'} =$ **ID-DAG**$(S', \mathbf{X}_Z, \mathcal{D}, G)$,    $\mathcal{D}' \sim H_{S'}(\mathbf{X}_Z)$
34:     **Return ID-DAG**$(\mathbf{Y}, \mathbf{X}, \mathcal{D}', G_{S', \overline{\mathbf{X}_Z}})$

---

network. Hence the operation of each recursive step can be summarized according to the following scheme: $i$) generate the appropriate dataset from joint samples to pass to the recursion, $ii$) make the recursive call and acquire the returned sampling networks, $iii$) update the global sampling network according to the output of the recursion. As we describe the algorithm, we will briefly motivate each case and describe how the operation fits into the above scheme.

**Base Cases**: We will describe the base cases of ID-DAG first. These are step 1 and step 6.

**Step 1** occurs when the current intervention set is empty. Here, the desired distribution we wish to sample from is $P(Y)$, and it suffices to only return a generative model for each variable in $Y$ from the dataset $\mathcal{D}$. This can be done by learning a generative model for each $Y_i \in Y$ conditioned on its parents according to the topological order of $G$ and wiring them to only depend on their parents.

**Step 6**: When we enter step 6, we are seeking to sample from $P_X(Y)$ where $Y$ is entirely in a single c-component $S$, and $X$ is disjoint from $S$. In this case, ID algorithm asserts that we can replace intervening on $X$ by conditioning on $X$. Similar to step 1, then, we can leverage samples from the dataset $\mathcal{D}$ to train a model for each variable $V_i$ in the c-component, conditioned only on ancestor nodes according to the topological order. Edges are added according to these conditioning variables.

Next we can consider the recursive cases, keeping in mind the schema that we need to $i$) alter the dataset, $ii$) acquire the sampling network from the recursion, $iii$) incorporate it into the global sampling network. We start with steps 2 and 3 as they only make a single recursive call to ID-DAG.

**Step 2:** We enter step 2 when there are variables $X_i \in X$ that are not ancestor of any $Y_i \in Y$. Intervening on such variables cannot affect $Y$ and we can safely drop them from the intervention set. We restrict the dataset $\mathcal{D}$ by only considering the variables that are ancestors of $Y$ according to

the topological order and stop considering the irrelevant $X_i's$. While bottom-up, we simply return the sampling network returned by the recursive call.

**Step 3:** In step 3, ID asserts that intervening on extra variables $W$ will have no influence on $Y$ assuming we already intervene on $X$. Here the dataset is the same, and we simply augment the interventional variables. We note that the choice of $W$ can be arbitrary, as discussed in Section 4.1.

**Step 4:** We enter this case when there are multiple c-components in the sub-graph $G \setminus X$ and one or more c-components are affected by $X$. Multiple recursive calls are made here, one for each c-component, with no alterations made to the dataset $\mathcal{D}$. Each call will return a sampling network for a c-component, but special care must be taken to merge these appropriately. The straightforward parallel to the ID algorithm would be to use Tian's factorization, where $P_X(V) = \prod_i P_{V \setminus S_i}(S_i)$. However each returned sampling network has one node per variable considered, depending only on its parents according to the global topological ordering. Here we make a call to the subroutine `ConstructDAG`, outlined in the appendix, to handle the correct edges to add to merge each of these graphs together. This step then returns a valid sampling network.

**Step 7:** This case occurs when i) $G \setminus X$ is a single c-component $S$, ii) all the variables in $Y$ are contained within a single c-component $S' \subset S$, but iii) the variables in $X$ can be partitioned into those that are contained within $S'$ and those that are not in $S'$. Letting $X_Z$ be those that are not contained within $S'$, ID asserts that evaluating $P_X(Y)$ is equivalent to evaluating $P'_{X \cap S'}(Y)$ where $P'$ is defined as $P_{X_Z}(V)$. Hence we need to generate a dataset $\mathcal{D}' \sim P_{X_Z}(V)$. This is handled by a recursive call to ID-DAG. Then we make the same recursive call as ID with the modification that we need to keep the variables $X_Z$ in the graph as conditioning variables. Please see appendix for more details on why this modification is needed.

**Sampling from ID-DAG:** Finally, after ID-DAG has returned a sampling network, we can sample from it as follows: specify values for any interventional variables in $X$, and choose arbitrary values for any variables $W$ that may have been added to the interventional set during step 3. Then proceed through the topological ordering of the DAG and call each generative model in turn. Of this joint sample, only return the variables in $Y$. We can show that this procedure is sound and complete:

**Theorem 4.1.** *(informal) ID-DAG is sound and complete for sampling from any identifiable $P_X(Y)$.*

**Conditional interventional sampling:** We provide the full conditional interventional sampling algorithm in Algorithm 5 and its soundness and completeness proof in Appendix C.

## 5 EXPERIMENTS

We apply our algorithms to two datasets. First, we consider a synthetic dataset using alterations to the thickness and color of MNIST images. Next we demonstrate that we can use Algorithm 1: ID-DAG to sample from an interventional distribution on a dataset involving real chest X-rays of COVID-19 patients. In both, we require the training of several neural networks, where full training details are deferred to the appendix. We remark that evaluation of correctness is challenging: The ground truth in these datasets is inaccessible and prior work was unable to compute interventional queries on high-dimensional data. Thus, our evaluations focus on the quality of neural network component and, in the case of MNIST, a surrogate ground-truth for a discrete version of the dataset.

For conditional sampling with high-dimensional data, we train a diffusion model using classifier-free guidance (Ho & Salimans, 2022). For conditional sampling of categorical data, we train a classifier using cross-entropy loss and apply temperature scaling as a means of calibration to ensure we are sampling in a calibrated fashion (Guo et al., 2017). Training details and code are in the appendix.

### 5.1 NAPKIN-MNIST DATASET

**Data Generation:** First we consider a synthetic dataset imbued over the napkin graph. Full data generation details are given in the appendix. We consider variables $W_1, W_2, X, Y$, where $W_1, X, Y$ are images derived from MNIST and $W_2$ is a discrete variable. We introduce latent confounders $C, T$, denoting color and thickness, where $C$ can be any of $\{red, greed, blue, yellow, magenta, cyan\}$, and $T$ can be any of $\{thin, regular, thick\}$. Data generation proceeds as follows: first we sample latent $C, T$ from the uniform distribution. We color and reweight a random digit from MNIST to

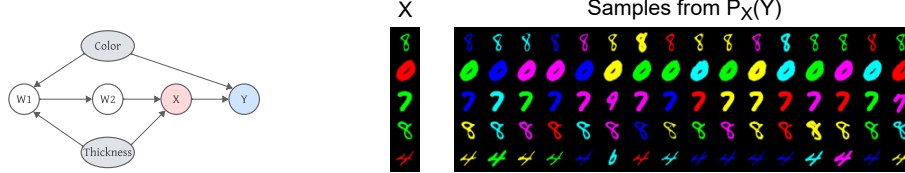

Figure 4: **Napkin MNIST:**. (Left) The causal graph for the Napkin-MNIST dataset. Thickness and Color are latent confounders. (Right): Samples from the interventional distribution $P_X(Y)$.

Table 1: **Color distributions from our generated interventional samples from** $P_X(Y)$, where $V.c$ refers to the color of variable $V$. $\hat{P}$ refers to our diffusion-based sampling mechanism, passed through a color classifier. All other values are the ground truth on the discrete Napkin-MINST. Our samples' colors closely reflect the true interventional distribution than other candidate distributions.

|  | Red | Green | Blue | Yellow | Magenta | Cyan |
|---|---|---|---|---|---|---|
| $\hat{\mathbf{P}}(\mathbf{Y.c} \mid \mathbf{Do}(\mathbf{X.c} = \mathbf{Red}))$ | **0.160** | **0.181** | **0.164** | **0.177** | **0.170** | **0.147** |
| $P(Y.c \mid Do(X.c = Red))$ | 0.167 | 0.167 | 0.167 | 0.167 | 0.167 | 0.167 |
| $P(Y.c \mid X.c = Red)$ | 0.276 | 0.276 | 0.276 | 0.057 | 0.057 | 0.057 |
| $\hat{\mathbf{P}}(\mathbf{Y.c} \mid \mathbf{Do}(\mathbf{X.c} = \mathbf{Green}))$ | **0.159** | **0.176** | **0.163** | **0.178** | **0.173** | **0.150** |
| $P(Y.c \mid Do(X.c = Green))$ | 0.167 | 0.167 | 0.167 | 0.167 | 0.167 | 0.167 |
| $P(Y.c \mid X.c = Green)$ | 0.057 | 0.057 | 0.057 | 0.276 | 0.276 | 0.276 |

form $W_1$. $W_2$ only keeps the digit value in $\{0 \dots 9\}$ of $W_1$ and a restriction of its color: if the color of $W_1$ is $red$, $green$, or $blue$, $W_2$'s color is $red$, and it is $green$ otherwise. $X$ then picks a random MNIST image of the same digit as $W_2$, is colored according to $W_2$'s color, and is reweighted according to the latent $T$. Then $Y$ is the same original MNIST image as $X$, reweighted according to $X$'s thickness and colored according to the latent $C$. Further, for every edge in the graph, we include a random noising process, where with $p = 0.1$, the information passed along is chosen uniformly randomly from the valid range. We are interested in sampling from the distribution $P_X(Y)$.

**Component Models:** The sampling net for the napkin graph is built by learning a model to sample from the distribution $P'(Y|X, W_2)$, where $P'(X, Y, W_2) := P_{W_2}(X, Y)$. Hence, given samples $\mathcal{D} \sim P$, we need to first generate a dataset $\mathcal{D}'$ for $W_2, X, Y$ from $P_{W_2}(X, Y)$, or equivalently just the $W_2, X, Y$ samples from $P(X, Y|W_1, W_2)P(W_1)$. Sampling from $P(X, Y|W_1, W_2)$ can be done by learning a conditional diffusion model trained on the observational distribution $P$. Samples from $P'$ can be done by sampling $W_1 \sim P(W_1)$, choosing an arbitrary $W_2$ and sampling from the trained diffusion model. On the new data $\mathcal{D}'$, we learn a diffusion model to sample from $P'(Y|X, W_2)$.

**Evaluation:** As the true ground truth interventional distribution is inaccessible, evaluation of our approach comes in two parts. First, we evaluate each of the trained neural nets for image fidelity and then we map our generated images to discrete variables where the ground truth is accessible. Examples from the trained diffusion models are given in Figure 4 (right). To evaluate the correctness of the sample, some preliminaries must be established. Note that each image in our data may be mapped to a discrete variable ($Digit, Color, Thickness$). Indeed, a discrete analogue of this dataset may be generated, for which exact likelihoods may be computed. Since we operate over images, we are only able to access these discrete properties through trained classifiers to identify digit, color, and thickness given an image. With classifiers in hand, we can estimate these properties of our sampled images from our learned $P_Y(X)$ and compare to the true (discrete) interventional and conditional distributions. We display these results for the color attribute in Table 1, and see that our sampling much more closely emulates the interventional distribution than the true conditional.

## 5.2 COVID X-RAY DATASET

**Data generation:** Next we apply our algorithm to a real dataset using chest X-rays on COVID-19 patients. Specifically, we download a collection of chest X-Rays (X) where each image has binary labels for the presence/absence of COVID-19 (C), and pneumonia (N) (Wang et al., 2020) [1]. We imbue the causal structure of the backdoor graph, where $C \to X$, $X \to N$, and there is a latent confounder affecting both $C$ and $N$ but not $X$. This may capture patient location, which might

---
[1] Labels are from https://github.com/giocoal/CXR-ACGAN-chest-xray-generator-covid19-pneumonia/

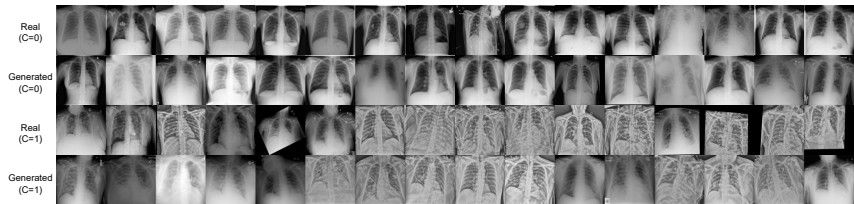

Figure 5: **Generated Covid XRay Images:** Generated chest XRay images from our diffusion model, separated by class and compared against real data.

affect the chance of getting COVID-19, and quality of healthcare affecting the right diagnosis. Since medical devices are standardized, location would not affect the X-ray image given COVID-19. We are interested in the interventional query $P_C(N)$: the treatment effect of COVID-19 on pneumonia.

**Component Models:** Applying ID-DAG to this graph requires access to two conditional distributions: $P(X|C)$ and $P(N|X,C)$. Since $X$ is a high-dimensional image, we train a conditional diffusion model to approximate the former. Since $N$ is a binary variable, we train a classifier that accepts $X, C$ and returns a Bernoulli distribution for $N$. The generated sampling network operates by sampling an $X$ given the interventional $C$, and then sampling an auxiliary $C' \sim P(C')$ and feeding $X, C'$ to the classifier for $P(N|X,C)$, finally sampling from this distribution.

**Evaluation:** Again we do not have access to the ground truth. Instead, we focus on the evaluation of each component model, and we also perform an ablation on our diffusion model. We first evaluate the image quality of the diffusion model approximating $P(X|C)$. We evaluate the FID of generated samples versus a held-out validation set of 10K X-ray images. When samples are generated with $C$ taken from the training distribution, we attain an FID score of $16.17$. We then evaluate the conditional generation by comparing class-separated FID evaluations and display these results in Table 2 (left). The classifier estimating $P(N|X,C)$ has an accuracy of 91.9% over validation set. We note that we apply temperature scaling (Guo et al., 2017) to calibrate our classifier, where the temperature parameter is trained over a random half of the validation set. Temperature scaling does not change the accuracy, but it does vastly improve the reliability metrics; see Appendix.

Finally we evaluate the query of interest $P_C(N)$. Since we cannot evaluate the ground truth, we consider our evaluated $P_C(N)$ versus an ablated version where we replace the diffusion sampling mechanism with $\hat{P}(X|C)$, where we randomly select an X-ray image from the held-out validation set. We also consider the query $P_C(N)$ if there were no latent confounders in the graph, in which case, the interventional query $P_C(N)$ is equal to $P(N|C)$. We display the results in Table 2 (right).

Table 2: **(Left) Class-conditional FID scores for generated Covid XRAY images** (lower is better). Generated $C = c$, means we sample from the diffusion model conditioned on $c$. Real $(C = c)$ refers to a held out test set of approximately 5k images, partitioned based on $C$-value. Low values on the diagonal and high values on the off-diagonal imply we are sampling correctly from conditional distributions. **(Right) Evalution of Interventional Distribution $P_C(N)$.** We evaluate the distributions $P_C(N = 1)$ for three cases for the Covid-XRAY dataset. Diffusion uses a learned diffusion model for $P(X|C)$, No Diffusion samples $P(X|C)$ empirically from a held out validation set, and no latent evaluates the conditional query assuming no latent confounders in the causal model.

| FID ($\downarrow$) | Real: $C = 0$ | Real: $C = 1$ |
| --- | --- | --- |
| Generated: $C = 0$ | 15.77 | 61.29 |
| Generated $C = 1$ | 101.76 | 23.34 |

| $P_c(N = 1)$ | $c = 0$ | $c = 1$ |
| --- | --- | --- |
| Diffusion | 0.622 | 0.834 |
| No Diffusion | 0.623 | 0.860 |
| No Latent | 0.406 | 0.951 |

## 6 CONCLUSION

In this paper, we propose an algorithm to sample from a conditional or unconditional high-dimensional interventional distributions. Our approach is able to leverage the state-of-the-art conditional generative models by showing that any identifiable causal effect estimand can be sampled from only via forward generative models. Our algorithm is sound and complete and, although we used diffusion models in our experiments, is agnostic to the specific choice of generative model.

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

# A  PSEUDO-CODES

## A.1  INTUITIVE EXPLANATION OF ORIGINAL ID-ALGORITHM(SHPITSER & PEARL, 2008)

---

**Algorithm 2** ID($\mathbf{y}, \mathbf{x}, P, G$)

---

1: **Input:** $\mathbf{y}, \mathbf{x}$ value assignments , distribution $P, G$.
2: **Output:** Expression for $P_{\mathbf{x}}(\mathbf{y})$ in terms of $P$ or Fail($F, F'$).
3: **if** $\mathbf{x} = \emptyset$ **then** {Step:1}
4:    **Return** $\sum_{\mathbf{v} \setminus \mathbf{y}} P(\mathbf{v})$
5: **if** $\mathbf{V} \setminus An(\mathbf{Y})_G \neq \emptyset$ **then** {Step:2}
6:    **Return ID**($\mathbf{y}, \mathbf{x} \cap An(\mathbf{Y})_G, \sum_{\mathbf{V} \setminus An(\mathbf{Y})_G} P, G_{An(\mathbf{Y})}$)
7: Let $\mathbf{W} = (\mathbf{V} \setminus \mathbf{X}) \setminus An(\mathbf{Y})_{G_{\overline{\mathbf{X}}}}$ {Step:3}
8: **if** $\mathbf{W} \neq \emptyset$ **then**
9:    **Return ID**($\mathbf{y}, \mathbf{x} \cup \mathbf{w}, P, G$)
10: **if** $C(G \setminus \mathbf{X}) = \{S_1, \ldots, S_k\}$ **then** {Step:4}
11:    **Return** $\sum_{\mathbf{v} \setminus (\mathbf{y} \cup \mathbf{x})} \prod_i ID(s_i, \mathbf{v} \setminus s_i, P, G$
12: **if** $C(G \setminus \mathbf{X}) = \{S\}$ **then**
13:    **if** $C(G) = \{G\}$ **then** {Step:5}
14:      **Return FAIL**($G, G \cap S$)
15:    **if** $S \in C(G)$ **then** {Step:6}
16:      **Return** $\sum_{s \setminus \mathbf{y}} \prod_{\{i | V_i \in S\}} P(v_i | v_\pi^{i-1})$
17:    **if** $\exists S'$ s.t. $S \subset S' \in C(G)$ **then** {Step:7}
18:      **Return ID**($\mathbf{y}, \mathbf{x} \cap S', P = \prod_{\{i | V_i \in S'\}} P(V_i | V_\pi^{(i-1)} \cap S', v_\pi^{(i-1)} \setminus S'), G_{S'}$)

---

**Algorithm 3** ConstructDAG($\{H_i\}_{\forall i}$)

---

1: **Input:** A set of DAGs containing trained models: $\{H_i\}_{\forall i}$.
2: **Output:** A DAG from $\{H_i\}_{\forall i}$
3: **for** $H_i \in \{H_i\}_{\forall i}$ **do**
4:   **for** $V_j \in H_i$ **do**
5:     **if** $V_j.f = \emptyset$ and $\exists V_k \in H_r, \forall r$ such that $V_j.name = V_k.name$ and $V_k.f \neq \emptyset$ **then**
6:       $V_j = V_k$
7: $H = \{H_i\}_{\forall i}$
8: **Return** $H$

---

**Algorithm 4** IDC($Y, X, Z, P, G$)

---

1: **Input:** $x, y, z$ value assignments, $P$ a probability distribution, $G$ a causal diagram (an I-map of $P$).
2: **Output:** Expression for $P_X(Y|Z)$ in terms of $P$ or Fail(F,F').
3: **if** $\exists \alpha \in Z$ such that $(Y \perp\!\!\!\perp \alpha | X, Z \setminus \{\alpha\})_{G_{\overline{X}, \underline{\alpha}}}$ **then**
4:    **return** IDC($Y, X \cup \{\alpha\}, Z \setminus \{\alpha\}, \mathcal{D}, G$)
5: **else**
6:    let $P' = $ ID($\mathbf{y} \cup \mathbf{z}, \mathbf{x}, P, G$)
7:    **return** $P' / \sum_{\mathbf{y}} P'$

---

---

**Algorithm 5** IDC-DAG($\mathbf{Y}, \mathbf{X}, \mathbf{Z}, \mathcal{D}, G$)

---

1: **Input:** target $Y$, intervention $X$ , conditioning set $Z$. training data $\mathcal{D}$, $G$.
2: **Output:** A DAG of trained model to sample from $P_{\mathbf{X}}(\mathbf{Y}|\mathbf{Z})$.
3: **if** $\exists \alpha \in Z$ such that $(Y \perp\!\!\!\perp \alpha | X, Z \setminus \{\alpha\})_{G_{\overline{X},\underline{\alpha}}}$ **then**
4:     **return** IDC-DAG($Y, X \cup \{\alpha\}, Z \setminus \{\alpha\}, \mathcal{D}, G$)
5: **else**
6:     $H_1 =$ ID-DAG($Y \cup Z, X, \mathcal{D}, G$)
7:     $\mathcal{D}' \sim H_1(X)$
8:     $H_2 = \emptyset$
9:     Add node $(X, \emptyset)$ and $(Z, \emptyset)$ to $H_2$
10:     Let $M_Y$ be a model trained on $\{Y, X, Z\} \sim \mathcal{D}'$ such that $M_Y(X, Z) \sim P'(Y|X, Z)$ i.e., $M_Y(X, Z) \sim P_X(Y|Z)$
11:     Add node $(Y, M_Y)$ to $H_2$
12:     Add edge $X \to Y$ and $Z \to Y$ to $H_2$
13:     **return** $H_2$

---

## B    APPENDIX: THEORY

Here we provide formal proofs for all theoretical claims made in the main paper, along with accompanying definitions and lemmas.

**Definition B.1.** A **conditional generative model** for a random variable $X \in V$ relative to distribution $P(V)$ is a function $f : \mathcal{P}a_X \to \mathcal{X}$ such that $f(pa_X) \sim P(X|pa_X), \forall pa \in \mathcal{P}a$, where $Pa$ is a subset of observed variables in $V$.

**Definition B.2.** A collection of conditional generative models for a set $V$ of variables is said to form a **sampling network** if the directed graph obtained by connecting each $X \in V$ to $Pa_X$ via incoming edges is acyclic. This graph is called the sampling network.

**Assumption B.3.** Each conditional generative model trained by ID-DAG correctly samples from the corresponding conditional distribution.

**Lemma B.4.** *Let $H$ be a sampling network for random variables $\{V_1, V_2, \ldots V_n\}$ formed by a collection of conditional generative models $f_i$ relative to $P_i$ for all $V_i$. Then the tuple $(V_1, V_2 \ldots V_n)$ obtained by sequentially evaluating each conditional generative model relative to the topological order of the sampling graph is a sample from the joint distribution $\Pi_i P_i(V_i | Pa_i)$.*

*Proof.* Without loss of generality, let $(V_1, V_2, \ldots, V_N)$ be a total order that is consistent with the topological ordering over the nodes in $G$. To attain a sample from the joint distribution, sample each $V_i$ in order. When sampling $V_j$, we will have already sampled $V_i$ for all $i < j$, which is a superset of $Pa_j$ by definition of topological orderings. $\qquad\square$

**Definition B.5.** We say that a sampling network $H$ is valid for an interventional distribution $p_X(Y)$ if the following conditions hold:

- Every variable $y \in Y$ has a conditional generative model in $H$.

- The only nodes in the sampling graph without conditional generative models are those in $X$.

- If the values $X = x$ are specified in $H$, then sampling $Y$ from $H$ is equivalent to sampling from $P_X(Y)$.

**Lemma B.6.** *Termination: Let $P_X(Y)$ be a query for causal graph $G = (V, E)$ and $\mathcal{D} \sim P(V)$. Then the recursions induced by $ID - DAG(Y, X, \mathcal{D}, G)$ terminate in either step 1, 5, or 6.*

*Proof.* Since ID is sound and complete, it is guaranteed to terminate in its base cases of 1, 5, or 6. Since the steps of ID-DAG exactly mirrors the steps of ID, with respect to recursions and the arguments $(Y, X, G)$ for every step except step 7, as long as ID-DAG does not enter step 7, it must terminate. In step 7, we make an extra call to ID-DAG of the form $ID - DAG(S', X_Z, \mathcal{D}, G)$. This follows exactly from Lemma 36 of Shpitser & Pearl (2008): by definition, there are no bidirected arcs from $X_Z$ to $S'$, so we proceed immediately to step 6, which is a base case. $\qquad\square$

**Lemma B.7.** *If a non-identifiable query is passed to our algorithm, ID-DAG, it will return* `FAIL`.

*Proof.* We implicitly run an identifiability check before running the ID-DAG presented in the pseudocode. If the query is not identifiable, ID will return fail, and thus so will ID-DAG. □

**Lemma B.8.** *ID-DAG Base case (step 1): Let $p(Y)$ be a query over causal graph $G = (V, E)$, and $\mathcal{D} \sim p(V)$ samples from the observational joint distribution over $V$. Then the graph returned by $ID - DAG(Y, \emptyset, \mathcal{D}, G)$ is a valid sampling network for $p(Y)$.*

*Proof.* Observe that $P(Y)$ may be factored as

$$P(Y) = \prod_{V_i \in Y} P(V_i | Y \cap V_\pi^{(i-1)}).$$

From Assumption 1, we may learn to sample from each conditional distribution in this product from joint samples. We can then add a node for each $V_i$, with its associated conditional distribution as a sampling mechanism. This produces a sampling graph that is a DAG, where each variable $V_i \in Y$ has a sampling mechanism. Since every node has a sampling mechanism and $X = \emptyset$, the third case for correctness holds. By the factorization and Assumption 1, sampling from this graph produces samples from $P(Y)$, hence this sampling graph is correct for $P(Y)$. □

**Lemma B.9.** *ID-DAG Base case (step 6): Let $P_X(Y)$ be an identifiable interventional query over a causal graph $G = (V, E)$, and that we have access to samples $\mathcal{D} \sim P(V)$ from the observational joint distribution over $V$. Suppose $ID - DAG(Y, X, \mathcal{D}, G)$ immediately enters step 6 and returns $H$. Then $H$ is a valid sampling network for $p_X(Y)$.*

*Proof.* In this case, both $ID - DAG(Y, X, \mathcal{D}, G)$ and $ID(Y, X, P, G)$ enter step 6. By the condition of step 6, $G \setminus X$ has only one c-component $\{S\}$, where $S \in C(G)$. Then the soundness of ID implies that

$$P_X(Y) = \sum_{S \setminus Y} \prod_{\{i | V_i \in S\}} P(V_i | V_\pi^{(i-1)})$$

where $\pi$ is a topological ordering for $G$. ID-DAG operates in this case by training, from joint samples $\mathcal{D}$, a model to correctly sample each $P(V_i \mid V_\pi^{(i-1)})$ term, i.e., we learn a conditional generative model $f_i(V_\pi^{(i-1)})$ which produces samples from $P(V_i \mid V_\pi^{(i-1)})$, which we can do according to Assumption 1. Then we construct a sampling network $H$ by adding a node $V_i$ with sampling mechanism $f_i$ for each $V_i \in S$. We add edges from $V_j \to V_i$ for each $V_j \in V_\pi^{(i-1)}$. Since every vertex in $G$ is either in $S$ or in $X$, every edge either connects to a previously constructed node or a variable in $X$. When specified values for $X$ and sampled according to topological order $\pi$, this sampling graph provides samples from the distribution $\prod_{\{i | V_i \in S\}} P(V_i | V_\pi^{(i-1)})$, i.e. $P_X(S)$. We assert the remaining conditions to show that this is correct for $P_X(Y)$: certainly this graph is a $DAG$ and every $v \in Y$ has a conditional generative model in $H$. By the conditions to enter step 6, if $C(G \setminus X) = \{S\}$, then $G = S \cup X$ and $S \cap X = \emptyset$. Then every node in $H$ is either in $S$ or is in $X$: hence the only nodes without sampling mechanisms are those in $X$ as desired. □

**Lemma B.10.** *Let $H$ be a sampling network produced by ID-DAG from an identifiable query $P_X(Y)$ over a graph $G$. If $G$ has the topological ordering $\pi$, then every edge in the sampling graph of $H$ adheres to the ordering $\pi$.*

*Proof.* We consider two factors: which edges are added, and with respect to which graphs. Since the only base cases ID-DAG enters are steps 1 and 6, the only edges added are consistent with the topological ordering $\pi$ for the graph that was supplied as an argument to these base case calls. The only graph modifications occur in steps 2 and 7, and these yield subgraphs of $G$. Thus the original topological ordering $\pi$ for graph $G$ is a valid topological ordering for each restriction of $G$. Therefore any edge added to $H$ is consistent with the global topological ordering $\Pi$. □

**Theorem B.11.** *ID-DAG Soundness: Let $P_X(Y)$ be an identifiable query given the causal graph $G = (V, E)$ and that we have access to joint samples $\mathcal{D} \sim P(V)$. Then the sampling network returned by $ID - DAG(Y, X, \mathcal{D}, G)$ correctly samples from $P_X(Y)$ under Assumption B.3.*

*Proof.* We proceed by structural induction. We start from the base cases, i.e., the steps that do not call ID-DAG again. ID-DAG only has three base cases: step 1 is the case when no variables are being intervened upon and is covered by Lemma B.8; step 6 is the other base case and is covered by Lemma B.9; step 5 is the non-identifiable case and since we assumed that $P_X(Y)$ is identifiable, due to Lemma B.7, ID-DAG never enters this step. The structure of our proof is as follows. By the assumption that $P_X(Y)$ is identifiable and due to Lemma B.6, its recursions must terminate in steps 1 or 6. Since we have already proven correctness for these cases, we use these as base cases for a structural induction. We prove that if ID-DAG enters any of step 2, 3, 4 or 7, under the inductive assumption that we have correct sampling graphs for the recursive calls, we can produce a correct overall sampling graph. The general flavor of these inductive steps adheres to the following recipe: i) determine the correct recursive call that ID algorithm makes; ii) argue that we can generate the correct dataset to be analogous to the distribution that ID uses in the recursion; iii) rely on the inductive assumption that the generated DAG from the recursion is correct.

We consider each recursive case separately. We start with step 2. Suppose ID-DAG$(Y, X, \mathcal{D}, G)$ enters step 2, then by the same conditions $ID(Y, X, P, G)$ enters step 2 as well. Hence the correct distribution to sample from is provided by ID step 2:

$$P_X(Y) = ID(Y, X \cap An(Y)_G, \sum_{V \setminus An(Y)_G} P(V), G_{An(Y)}).$$

Following our recipe, we need to generate the dataset sampled from $\sum_{V \setminus An(Y)_G} P$, generated only with samples from $D \sim p(V)$. This amounts to converting joint samples $\mathcal{D}$ to marginal samples $\mathcal{D}'$. We do this by dropping all variables downstream of $Y$ (in the graph $G$) from the dataset $\mathcal{D}$, thereby attaining samples from the joint distribution $\sum_{V \setminus An(Y)_G} P(V)$. Then we can generate the sampling network from ID-DAG$(Y, X \cap An(Y)_G, \mathcal{D}', G_{An(Y)})$ by the inductive assumption and simply return it.

Next, we consider step 3. Suppose ID-DAG$(Y, X, \mathcal{D}, G)$ enters step 3. Then by the same conditions, $ID(Y, X, P, G)$ enters step 3, and the correct distribution to sample from is provided from ID step 3 as

$$p_X(Y) = ID(Y, X \cup W, P, G)$$

where $W := (V \setminus X) \setminus An(Y)_{G_{\bar{X}}}$. Since the distribution passed to the recursive call is $P$, we can simply return the sampling graph generated by ID-DAG$(Y, X \cup W, D, G)$, which we know is correct for $P_{X \cup W}(Y)$ by the inductive assumption. While we do need to specify a sampling mechanism for $W$ to satisfy our definition of a valid sampling network, this can be chosen arbitrarily, say $W \sim P(W)$. Note that these freely-chosen distributions will not affect the final samples as they will cancel out due to conditioning on these variables.

Next we consider step 4. Suppose ID-DAG$(Y, X, \mathcal{D}, G)$ enters step 4. Then by the same conditions, $ID(Y, X, P, G)$ enters step 4 and the correct distribution to sample from is provided from ID step 4 as:

$$\sum_{V \setminus (y \cup x)} \prod_i ID(s_i, v \setminus s_i, P, G)$$

where $S_i$ are the c-components of $G \setminus X$, i.e., elements of $C(G \setminus X)$. By the inductive assumption, we can sample from each term in the product by ID-DAG$(s_i, v \setminus s_i, D, G)$. However, recall the output of ID-DAG: ID-DAG returns a 'headless' sampling network as follows: ID-DAG$(Y, X, \mathcal{D}, G)$ is a sampling network, i.e., a collection of conditional generative models where for each variable in $G$ and every variable except those in $X$ have a specified conditional generative model. To sample from this sampling network, values for $X$ must first be specified. In the step 4 case, the values $V \setminus s_i$ need to be provided to sample values for $s_i$, and similarly for $i \neq j$, values $V \setminus s_j$ are needed to sample values for $s_j$. Since $s_i \subseteq (V \setminus s_j)$ and $s_j \subseteq (V \setminus s_i)$, this might lead to cycles in the corresponding directed graph of the sampling network obtained by combining each conditional generative model for each c-component. Thus, it does not suffice to sample from each c-component separately. Hence if $H_i$ is the correct sampling network corresponding to ID-DAG$(S_i, V \setminus S_i, P, G)$ by definition, for each node $v_i \in S_i$, $v_i$ has a conditional generative model in $H_i$. By Lemma B.10, each edge in $H_i$ adheres to the topological ordering $\Pi$ for $G$. Hence we apply ConstructDAG to construct a graph $H$ from $\{H_i\}_i$ which also adheres to the original topological ordering $\pi$. Thus $H$ is a DAG. Since every node in $G \setminus X$ has a conditional generative model in some $H_i$, no $v \in X$ has a conditional generative model in any $H_i$, the only nodes in $H$ without conditional generative models are those

in $X$. Finally, since each node in $H$ samples the correct conditional distribution by the inductive assumption, $H$ samples from the distribution $P_X(Y)$. The sum $\sum_{V \setminus (y \cup x)}$ can be safely be ignored, because samples from the joint can be marginalized to attain samples from marginals. Hence $H$ is correct for $P_X(Y)$.

Step 5 can never happen by the assumption that $P_X(Y)$ is identifiable, and step 6 has already been covered as a base case. The only step remaining is step 7.

Supposing ID-DAG$(Y, X, \mathcal{D}, G)$ enters step 7, then by the same conditions $ID(Y, X, P, G)$ enters step 7. Then by assumption, $C(G \setminus X) = \{S\}$ and there exists a confounding component $S' \in C(G)$ such that $S \subset S'$. The correct distribution to sample from is provided from ID step 7 as

$$p_X(Y) = ID(Y, X \cup S', P', G_{S'})$$

where

$$P' := \prod_{\{i|V_i \in S'\}} p(V_i | V_\pi^{(i-1)} \cup S', v_\pi^{(i-1)} \setminus S').$$

Examining ID algorithm more closely, if we enter step 7 during ID, the interventional set $X$ is partitioned into two components: $X \cap S'$ and $X_Z := X \setminus S'$. From Lemmas 33 and 37 of Shpitser & Pearl (2008), in the event we enter step 7, $P_X(Y)$ is equivalent to $P'_{X \cap S'}(Y)$ where $P'(V) = P_{X_Z}(V)$. Hence in order to sample correctly, we need to do two things: first we need to alter our samples $\mathcal{D} \sim P$ to samples from $\mathcal{D}' \sim P_{X_Z}(V)$, and then we need to recurse on the query $P'_{X \cap S'}(Y)$ over the graph $G_{S'}$. Generating samples $\mathcal{D}' \sim P_{X_Z}(V)$ can further be reduced to samples only from $X \cup Y$, which we note is equivalent to $S' \cup X_Z$. Hence the dataset $\mathcal{D}'$ need only be supported over $S'$. Therefore we need to generate a dataset from $\mathcal{D}'$ via ID-DAG$(S', X_Z, \mathcal{D}, G)$. This is attainable via the inductive assumption and Lemma B.6. The only divergence from ID during generation of $\mathcal{D}'$ is that ID presumes pre-specified values for $X$, where we train a sampling mechanism that is agnostic a priori to the specific choice of $X$ and hence $X_Z$. To sidestep this issue, we generate a dataset with all possible values of $X_Z$ and be sure to record the values of $X_Z$ in the dataset $\mathcal{D}'$. Next, we need to map the recursive call $ID(Y, X \cap S', P', G)$ to ID-DAG. Since instead of passing the distribution $P'$, we pass the dataset $\mathcal{D}'$, which does not have specified values for $X_Z$, we need to pass the interventional values for $X_Z$ into this recursive call. Hence, instead of intervening in the recursion on $X \cap S'$, we intervene on $X$ or equivalently $(X \cap S') \cup X_Z$. Additionally, since we are intervening on $X \cap S'$ and $X_Z$, we need to ensure $X_Z$ is included in the restricted graph, hence we replace $G_{S'}$ with $G_{S', \overline{X_Z}}$. By the inductive assumption, we can generate a correct sampling graph from the call ID-DAG$(Y, (X \cap S') \cup X_Z, \mathcal{D}', G_{S', \overline{X_Z}})$, and hence the returned sampling graph is correct for $P_X(Y)$.

Since we have shown that every recursion of ID-DAG ultimately terminates in a base case, that all the base cases provide correct sampling graphs, and that correct sampling graphs can be constructed in each step assuming the recursive calls are correct, we conclude that ID-DAG returns the correct sampling graph for $P_X(Y)$.

$\square$

## C APPENDIX: CONDITIONAL INTERVENTIONAL SAMPLING

**Conditional sampling:** Given a conditional causal query $P_X(Y|Z)$, we sample from this conditional interventional query by calling Algorithm 5: IDC-DAG. This function finds the maximal set $\alpha \subset Z$ such that we can apply rule-2 and move $\alpha$ from conditioning set $Z$ and add it to intervention set $X$. Precisely, $P_X(Y|Z) = P_{x \cup \alpha}(Y|Z \setminus \alpha) = \frac{P_{x \cup \alpha}(Y, Z \setminus \alpha)}{P_{x \cup \alpha}(Z \setminus \alpha)}$. Next, Algorithm 1: $ID - DAG(.)$ is called to obtain the sampling network that can sample from the interventional joint distribution $P_{X \cup \alpha}(Y, Z \setminus \alpha)$. We use the sampling network to generate samples $\mathcal{D}'$ through feed-forward. A new conditional model $M_Y$ is trained on $\mathcal{D}'$ that takes $Z \setminus \alpha$ and $X \cup \alpha$ as input and outputs $Y$. Finally, we generate new samples with $M_Y$ by feeding input values such that $Y \sim P_{X \cup \alpha}(Y, Z \setminus \alpha)$ i.e, $Y \sim P_X(Y|Z)$.

**Theorem C.1** (Shpitser & Pearl (2008)). *For any $G$ and any conditional effect $P_{\mathbf{X}}(\mathbf{Y}|\mathbf{W})$ there exists a unique maximal set $\mathbf{Z} = \{Z \in \mathbf{W} | P_{\mathbf{X}}(\mathbf{Y}|\mathbf{W}) = P_{\mathbf{X}, \mathbf{z}}(\mathbf{Y}|\mathbf{W} \setminus Z)\}$ such that rule 2 applies to $\mathbf{Z}$ in $G$ for $P_{\mathbf{X}}(\mathbf{Y}|\mathbf{W})$. In other words, $P_{\mathbf{X}}(\mathbf{Y}|\mathbf{W}) = P_{\mathbf{X}, \mathbf{z}}(\mathbf{Y}|\mathbf{W} \setminus \mathbf{Z})$.*

**Theorem C.2** (Shpitser & Pearl (2008)). *Let $P_{\mathbf{X}}(\mathbf{Y}|\mathbf{W})$ be such that every $W \in \mathbf{W}$ has a backdoor path to $Y$ in $G \setminus \mathbf{X}$ given $\mathbf{W} \setminus \{W\}$. Then $P_{\mathbf{X}}(\mathbf{Y}|\mathbf{W})$ is identifiable in $G$ if and only if $P_{\mathbf{X}}(\mathbf{Y}, \mathbf{W})$ is identifiable in $G$.*

**Theorem C.3.** *IDC-DAG Soundness: Let $P_X(Y|Z)$ be an identifiable query given the causal graph $G = (V, E)$ and that we have access to joint samples $\mathcal{D} \sim P(V)$. Then the sampling network returned by IDC-DAG$(Y, X, Z, \mathcal{D}, G)$ correctly samples from $P_X(Y|Z)$ under Assumption B.3*

*Proof.* The IDC algorithm is sound and complete based on Theorem C.1 and Theorem C.2. For sampling from the conditional interventional query, we follow the same steps as the IDC algorithm in Algorithm 5: IDC-DAG. Therefore, IDC-DAG is sound and complete. □

# D    TECHNICAL NOVELTIES OF THE ID-DAG ALGORITHM

---
ConstructDAG($\{H_i\}_{\forall i}$)

---
1: **Input:** A set of DAGs containing trained models: $\{H_i\}_{\forall i}$.
2: **Output:** A DAG from $\{H_i\}_{\forall i}$
3: **for** $H_i \in \{H_i\}_{\forall i}$ **do**
4:     **for** $V_j \in H_i$ **do**
5:         **if** $V_j.f = \emptyset$ and $\exists V_k \in H_r, \forall r$ such that $V_j.name = V_k.name$ and $V_k.f \neq \emptyset$ **then**
6:             $V_j = V_k$
7: $H = \{H_i\}_{\forall i}$
8: **Return** $H$

---

## D.1    SUB-PROCEDURE: ConstructDAG()

Each sampling network $H_i$ is a set of conditional trained models connected with each other according to a directed acyclic graph. Any node $V_j$ in the sampling network, represents a variable name $V_j.name$ and its corresponding trained model $V_j.f$. This sub-procedure takes multiple sampling networks $\{H_i\}_{\forall i}$ as input and combines them to construct a larger consistent sampling network $H$. We iterate over the nodes $V_j$ of each sampling network $H_i$ (lines 3-4). If $V_j$ does not have any corresponding trained model associated with it, it must have its conditional model located in some other sampling network $H_r$ as node $V_k$. When we find that node $V_k$ in another sampling network $H_r$ (line 5), we combine $H_i$ and $H_r$ by combining $V_j$ and $V_k$ since $V_j$ and $V_k$ are the same variable (line 6). In this manner, we combine all sampling network to construct one single network $H$.

## D.2    STEP 4 OF THE ID-DAG ALGORITHM

| ID algorithm Step 4 | ID-DAG algorithm Step 4 |
|---|---|
| 1: **if** $C(G \setminus \mathbf{X}) = \{S_1, \ldots, S_k\}$ **then** {Step:4}
2:     **Return** $\sum_{\mathbf{v} \setminus (\mathbf{y} \cup \mathbf{x})} \prod_i ID(s_i, \mathbf{v} \setminus s_i, P, G)$ | 1: **if** $C(G \setminus \mathbf{X}) = \{S_1, \ldots, S_k\}$ **then** {Step 4}
2:     **for** each $S_i \in C(G \setminus \mathbf{X}) = \{S_1, \ldots, S_k\}$ **do**
3:         $H_i$=**ID-DAG**$(S_i, \mathbf{X} = \mathbf{V} \setminus S_i, \mathcal{D}, G)$
4:     **Return** ConstructDAG$(\{H_i\}_{\forall i})$ |

Step 4 of the ID algorithm performs Tian's factorization, i.e, splits the variables in $G \setminus \mathbf{X}$ into multiple c-components and estimates $P_{\mathbf{v} \setminus s_i}(s_i)$ recursively. Finally, multiplies these factors and marginalizes all variables except $\mathbf{y}$ and $\mathbf{x}$. To sample from the corresponding causal query $P_{\mathbf{x}}(\mathbf{Y})$, one might feel tempted to sample $S_i$ from each $P_{\mathbf{v} \setminus s_i}(s_i)$ in topological order and then combine them. However, this might lead to cyclic situation. For example, consider the causal graph in Figure 6. Here, according to the ID algorithm step 4, $P_x(y) = \sum_{w_1, w_2} P_{w_1}(x, w_2) * P_{x, w_2}(w_1, y)$. To sample $X, W_2 \sim P_{w_1}(x, w_2)$ we need $W_1$ as input which has to be sampled from $P_{x, w_2}(w_1, y)$. But to sample $W_1 \sim P_{x, w_2}(w_1, y)$, we need $X, W_2$ as input which has to be sampled from $P_{w_1}(x, w_2)$. Therefore, no order helps to sample all $X, W_1, W_2, Y$ consistently. ID-DAG solves this cyclic issue

by avoiding direct sampling from the c-components. Rather ID-DAG builds a sampling network $H_i$ consists of trained models for each c-component $S_i$. After that, sub-procedure $\text{ConstructDAG}$ is called to merge the sampling network found from each c-component to build a one single sampling network. We use this single sampling network to sample $[S_1, \ldots, S_k] \sim P_{V \setminus S_1}(S_1) * \ldots P_{V \setminus S_k}(S_k)$. ID-DAG drops all variables from $[S_1, \ldots, S_k]$ except $Y$ and these will be the samples from $P_x(Y)$.

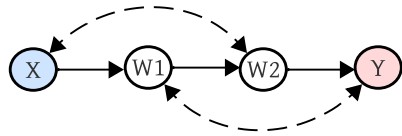

Figure 6: Cyclic graph and failure case for topological order sampling

## D.3 STEP 7 OF THE ID-DAG ALGORITHM

| ID algorithm Step 7 |
| --- |
| 1: **if** $\exists S'$ s.t. $S \subset S' \in C(G)$ **then** {Step:7} |
| 2:     **Return ID**$(\mathbf{y}, \mathbf{x} \cap S', P = $ |
|     $\prod_{\{i|V_i \in S'\}} P(V_i | V_\pi^{(i-1)} \cap S', v_\pi^{(i-1)} \setminus S'),$ |
|     $G_{S'})$ |

| ID-DAG algorithm Step 7 |
| --- |
| 1: **if** $\exists S'$ s.t. $S \subset S' \in C(G)$ **then** {Step 7} |
| 2:     $\mathbf{X}_Z = \mathbf{X} \setminus S',$ |
| 3:     $H_{S'} = $ **ID-DAG**$(S', \mathbf{X}_Z, \mathcal{D}, G)$ |
| 4:     $\mathcal{D}' \sim H_{S'}(\mathbf{X}_Z)$ |
| 5:     **Return ID-DAG**$(\mathbf{Y}, \mathbf{X}, \mathcal{D}', G_{S', \overline{\mathbf{X}_Z}})$ |

Here we show the differences between step 7 of the ID algorithm and the ID-DAG algorithm which represent the technical novelty of our method. This case occurs when i) $G \setminus X$ is a single c-component $S$, ii) all the variables in $Y$ are contained within a single c-component $S' \subset S$, but iii) the variables in $X$ can be partitioned into those that are contained within $S'$ and those that are not in $S'$. ID algorithm uses $P = \prod_{\{i|V_i \in S'\}} P(V_i|V_\pi^{(i-1)} \cap S', v_\pi^{(i-1)} \setminus S')$, $\mathbf{x} = \mathbf{x} \cap S'$ and $G = G_{S'}$ as arguments values for the next recursive call.

**ID-DAG modification with respect to the dataset $\mathcal{D}$:** This new P is actually an interventional distribution, $P_{X \setminus S'}(S')$ which is identifiable from obs data with the mentioned expression. It is not straightforward to see the role of the new distribution argument value used in the ID algorithm. Let $\mathbf{X}_Z = \mathbf{X} \setminus S'$, i.e, the subset of intervention set $\mathbf{X}$ that is not contained within $S'$. ID-algorithm under the hood using the fact that evaluating $P_X(Y)$ where $P = P(V)$ is equivalent to evaluating $P'_{X \cap S'}(Y)$ where $P' = P_{X_Z}(V)$.

Therefore, to train the conditional models in the subsequent steps in the recursion, ID-DAG generates an interventional dataset from $P_{X_Z}(V)$. To do that ID-DAG first obtains the sampling network with a **ID-DAG**$(S', \mathbf{X}_Z, \mathcal{D}, G)$ call. Then the algorithm uses this sampling network to obtain the interventional dataset $\mathcal{D}' \sim P_{X_Z}(V)$ which is used in the next recursive call.

**ID-DAG modification with respect to the graph $G$:** The graph argument ID algorithm sends to the next recursive call is $G_{S'}$, i.e, it removes the intervened variables $X_Z$ from the graph G. This works well for the ID algorithm because all variables values are assumed at the beginning and it estimates the corresponding probability. However, this does not work if our aim is sampling. Since what values of $X_Z$ we are using here depend on the values $X_Z$ has taken in another c-component, we consider all possible values of $X_Z$ while training all conditional models in any level of the recursion for a fixed c-component. Thus, we do not remove $X_Z$ from the causal graph but remove the incoming edges since we intervened on them. As a result, when ever we train any conditional model in a deeper level of the recursion, it will be used as an input condition. This is a very important step to sample from $P_X(Y)$ consistently.

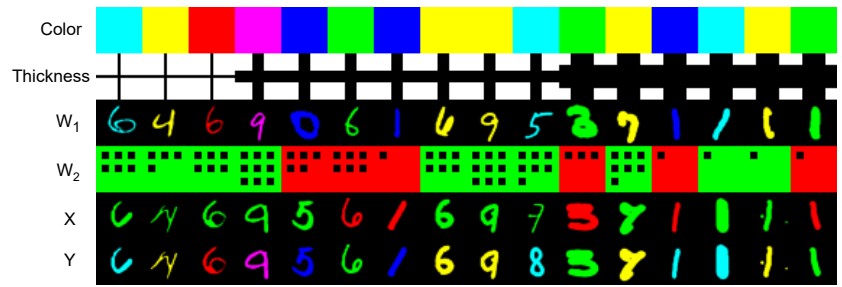

Figure 7: **Joint samples from the Napkin-MNIST dataset:** Samples from the Napkin-MNIST dataset are visualized as columns above. The first row indicates the latent variable `color`, the second row indicates the latent variable `thickness`, and the row labeled $W_2$ is a discrete variable holding a (`color`, `digit`), where digit is represented as the number of dots. Notice that the noising process sometimes causes information to not be passed to children.

# E    APPENDIX: EXPERIMENTAL DETAILS

## E.1    NAPKIN-MNIST DATASET

Here we describe the data-generation procedure, and training setup for the Napkin-MNIST experiment in full detail.

### E.1.1    DATA GENERATION PROCEDURE: DISCRETE CASE

As a warm-up, we outline the generation for the Napkin-MNIST dataset in a low-dimensional setting. When we consider in the next section the high-dimensional case, we simply replace some of these discrete variables with MNIST images which can be mapped back into this low-dimensional case.

We start by enumerating the joint distribution and the support of each marginal variable. First lets define the sets

- COLORS := {red, green, blue, yellow, magenta, cyan}.
- RG_COLORS := {red, green}.
- THICKNESSES := {thin, regular, thick}.
- DIGITS := {0, ..., 9}.

And then the definitions and support of each of the variables in our distribution:

- (Latent) Color ∈ COLORS.
- (Latent) Thickness ∈ THICKNESSES.
- $W_1 \in$ DIGITS × COLORS × THICKNESSES
- $W_2 \in$ DIGITS × RG_COLORS.
- $X \in$ DIGITS × COLORS × THICKNESSES
- $Y \in$ DIGITS × COLORS × THICKNESSES

Now we describe the full data generation procedure. A key hyperparameter is a noise-probability $p$. This defines the probability that any variable flips to a uniform probability. To ease notation, we define the function $\eta_p(v, S)$ defined as

$$\eta_p(v, S) := \begin{cases} v & \text{with probability } 1 - p \\ U(S) & \text{otherwise} \end{cases}$$

and we define the mapping $R : $ COLORS $\rightarrow$ RESTRICTED_COLORS as

$$R(c) := \begin{cases} \text{red} & \text{if } c \in \{\text{red, green, blue}\} \\ \text{green} & \text{otherwise} \end{cases}$$

Where $U(S)$ means a uniformly random choice of $S$. Then our data generation procedure follows the following steps:

- `Color` $:= U(\text{COLORS})$
- `Thickness` $:= U(\text{THICKNESSES})$
- $W_1 := \big(U(\text{DIGITS}), \quad \eta_p(\text{Color}, \text{COLORS}), \quad \eta_p(\text{Thickness}, \text{THICKNESSES})\big)$
- $W_2 := \big(\eta_p(W_{1 \cdot digit}, \text{DIGITS}), \quad \eta_p(R(W_{1 \cdot color}, \text{RG\_COLORS}))\big)$
- $X := \big(\eta_p(W_{2 \cdot digit}, \text{DIGITS}), \quad \eta_p(W_{2 \cdot color}, \text{RG\_COLORS}), \quad \eta_p(\text{Thickness}, \text{THICKNESSES})\big)$
- $Y := \big(\eta_p(X_{\cdot digit}, \text{DIGITS}), \quad \eta_p(\text{Color}, \text{COLORS}), \quad \eta_p(X_{\cdot thickness}, \text{THICKNESSES})\big)$

It is easy to verify that this describes the Napkin graph, as each only `Color, Thickness` are latent and each variable only depends on its parents in the SCM.

Secondly, observe that this structural causal model is separable with respect to digits, colors, and thicknesses. Since each digit only depends on parent *digits*, each color only depends on parent *colors*, and each thickness depends only on parent *thicknesses*, these can all be considered separately.

Further, because this distribution is only supported over discrete variables, exact likelihoods can be computed for any conditional query. This is much more easily done programmatically, however, and we provide code in the attached codebase to do just that. We will claim without proof that in the case of thicknesses and digits, $P_Y(X) = P(Y|X)$. However in the case of colors, $P_Y(X) \neq P(Y|X)$. Hence we consider this case in the evaluations in the experiments section.

### E.1.2 DATA GENERATION PROCEDURE: HIGH-DIMENSIONAL CASE

The high-dimensional case follows the discrete case of the Napkin-MNIST dataset, with a few key changes. Namely, $W_1, X$, and $Y$ are MNIST images that have been colored and thickened. We explicitly outline these changes:

- $W_1$: A random MNIST image of the provided digit is used, then colored and thickened accordingly (noisy from latents).
- $W_2$ : This is a discrete variable, only encoding the (noised) digit and (noised) restricted color of $W_1$.
- $X$: This is a random MNIST image of the (noised) digit obtained from $W_2$, then colored with the (noised) restricted color from $W_2$ and thickened according to the (noised) latent thickness.
- $Y$: This is the *same* base image of $X$, unless the noising procedure calls for a change in digit, then a random MNIST image of the specified image is used. The (noisy) color is obtained from the latent distribution, and the (noisy) thickness is obtained from $X$.

To color the images, we convert each 1-channel MNIST image into a 3-channel MNIST image, and populate the necessary channels to generate these colors. Note that in RGB images: if only the RG channels are active, the image is yellow; if only the RB channels are active, the image is magenta; if only the BG channels are active, the image is cyan. To thicken the images, we use the MorphoMNIST package[2]. Operationally, we generate a base dataset for our experiments of size equivalent to the original MNIST dataset. That is, the training set has a size of 60K, and the test set has a size of 10K. Because we have access to the latents during the data generation procedure, we are able to train classifiers for each variable to identify their digit, color and thickness. We use a simple convolutional network architecture for each of these cases and achieve accuracy upwards of 95% in each case.

### E.1.3 DIFFUSION TRAINING DETAILS

We train two diffusion models during our sampling procedure, and we discuss each of them in turn.

---

[2]https://github.com/dccastro/Morpho-MNIST/

To train a model to sample from $P(X, Y | W_1, W_2)$, we train a *single* diffusion model over the joint $(X, Y)$ distribution, i.e., 6 channels. We train a standard UNet architecture where we follow the conditioning scheme of classifier-free guidance. That is, we insert at every layer an embedding of the $W_1$ (image) and $W_2$ (2-dimensional discrete variable). To embed the $W_1$ image, we use the base of a 2-layer convolutional neural network for MNIST images, and to embed the $W_1$ we use a standard one-hot embedding for each of the variables. All three embeddings are concatenated and mixed through a 2-layer fully connected network to reach a final embedding dimension of 64. Batch sizes of 256 are used everywhere. Training is performed for 1000 epochs, which takes roughly 9 hours on 2 A100 GPU's. Sampling is performed using DDIM over 100 timesteps, with a conditioning weight of $w = 1$ (true conditional sampling) and noise $\sigma = 0.3$.

To train a model to sample $Y$ from the generated dataset $(W_2, X, Y)$, we follow an identical scheme. Perhaps the most correct thing is to train a single diffusion model for each choice of $W_2$ in our synthetic dataset, however we argue our schema still produces correct samples because: 1) $W_2$ can be arbitrarily chosen, and thus should not affect $Y$, 2) we argue that the model fidelity benefits from weight sharing across multiple choices of $W_2$, 3) the model is only ever called with a specified value of $W_2$ so we always condition on this $W_2$.

### E.1.4 EXTRA EVALUATIONS

In addition to the evaluations presented in the main paper, we can further perform evaluations on the component models necessary to sample $P_X(Y)$.

$P(X, Y | W_1, W_2)$: We can evaluate the model approximating samples from $P(X, Y | W_1, W_2)$ on a deeper level than just visual inspection as provided in the main paper. In particular, assuming access to good classifiers that can predict the digit, color, and thickness of an MNIST image, we can compare properties of the generated images with respect to the ground truth in the discrete case. For example, assuming we have hyperparameter of random noise equal to $p$, we can compute the following quantities analytically on the discrete dataset as:

- $P[X_d = W_{2d}] = 1 - p + \frac{p}{10}$
- $P[X_{\cdot c} = W_{2c}] = 1 - p + \frac{p}{10}$
- $P[X_t = W_{1t}] = (1 - p + \frac{p}{3})^2 + \left(\frac{p}{3}\right)^2 \cdot 2$
- $P[Y_d = X_d] = 1 - p + \frac{p}{10}$
- $P[Y_c = W_{1c}] = (1 - p + \frac{p}{6})^2 + big(\frac{p}{6}\right)^2 \cdot 5$
- $P[Y_t = X_t] = 1 - p + \frac{p}{3}$

where $V_d, V_c, V_t$ refer to the digit, color, and thickness attributes respectively. These calculations follow from two formulas. In a discrete distribution with support $S$ and $|S| = K$:

- $P[\eta_p(z, S) = z] = 1 - p + \frac{p}{K}$
- $P[\eta_p(z, S) = \eta_p(z, S)] = (1 - p + \frac{p}{K})^2 + \left(\frac{p}{K}\right)^2 \cdot (K - 1)$

where in the second equation, it is assumed that $\eta_p(\cdot, \cdot)$ are two independent noising procedures.

Then to evaluate, we can 1) consider a large corpus of joint data, 2) run each of $W_1, X, Y$ through a classifier for digit, color, and thickness, 3) evaluate the empirical estimate of each desired probability. We present these results for the synthetic dataset $D_{synth}$ sampled from the diffusion model approximating $P(X, Y | W_1, W_2)$, a dataset $D_{orig}$ generated according to the data generation procedure, and $P_{true}$ the true analytical probabilities. These results are displayed in the Table 3.

### E.1.5 BASELINE COMPARISON

Here we compare our performance with two baselines and provide the results below: We provide a short description of the ground truth first. We have the napkin causal graph:

$$W_1 \to W_2 \to X \to Y; X \leftrightarrow W_1 \leftrightarrow Y$$

Table 3: **Evaluations on the Napkin-MNIST generated dataset**. $V.d, V.c, V.t$ refer to the digit, color and thickness respectively of variable $V$. The first column is with respect to samples generated from diffusion model $\hat{P}(X, Y \mid W_1, W_2)$, Image Data is the dataset used to train $\hat{P}$, Discrete Data is the empirical distribution according to a discrete Napkin-MNIST, and the ground truth is analytically computed. Ideally all values should be equal across a row. While our synthetic dataset generated from $\hat{P}$ is not a perfect representation, it is quite close in all attributes except thickness. This is because the classifier for thickness has some inherent error in it, as evidenced by the mismatch between the base data and ground truth in the thickness rows.

| | $\hat{P}(Y, X \mid W_1, W_2)$ | Image Data | Discrete Data | Ground Truth |
|---|---|---|---|---|
| $P[X.d = W_2.d]$ | 0.931 | 0.895 | 0.909 | 0.910 |
| $P[X.c = W_2.c]$ | 0.964 | 0.950 | 0.950 | 0.950 |
| $P[X.t = W_1.t]$ | 0.683 | 0.776 | 0.879 | 0.873 |
| $P[Y.d = X.d]$ | 0.927 | 0.895 | 0.909 | 0.910 |
| $P[Y.c = W_1.c]$ | 0.847 | 0.841 | 0.841 | 0.842 |
| $P[Y.t = X.t]$ | 0.830 | 0.851 | 0.933 | 0.933 |

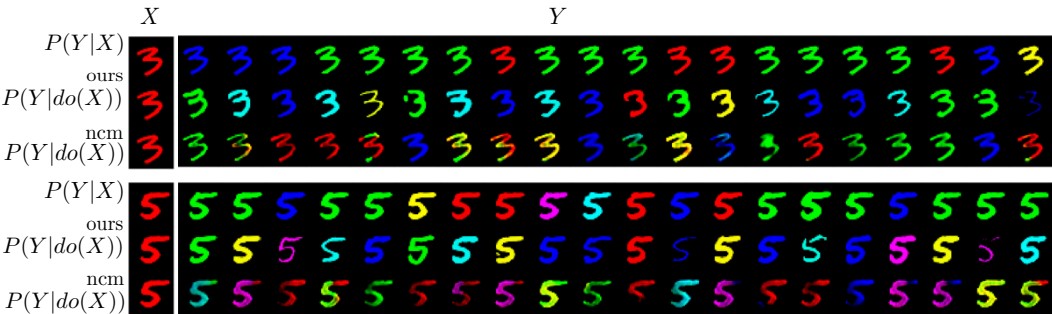

Figure 8: Performance comparison with baselines

**Dataset:** $W_1, X$ and $Y$ are digit images and $W_2$ are discrete properties (thickness and color) of $W_1$. Each digit image can take a value from [0,9] and color from [Red, Green, Blue, Yellow, Magenta, Cyan].

**Baselines**: We compared our algorithm with two baselines: Baseline 1: A classifier-free diffusion model that samples from the conditional distribution: $Y \sim P(Y|X)$. Baseline 2: The NCM algorithm Xia et al. (2021) that samples from the interventional distribution $P(Y|do(X))$.

**Algorithm execution:** To easily understand the distinction among different methods, we chose two images i) of digit 3 and ii) of digit 5, both colored red as the intervention value for $X$. Then we used these images to sample from the corresponding distribution $P(Y|X)$ and $P(Y|do(X))$ distributions using different methods. Although the implementation of these methods is different, we considered the performance of each method after running them for 300 epochs. The NCM algorithm took around 50 hours to complete while our algorithm took approximately 16 hours to complete.

**Results:**

i) Image quality: First, we discuss the image quality of the generated samples from each approach that we provided in Figure 8. The conditional model (row 1, row 4) and our algorithm (row 2, row 5) both generate high-quality images of digit 3 and digit 5 with a specific color from the six possible colors. Whereas, the colors in generated images from the NCM algorithm (row 3, row 6) can not remain at any specific value and get blended (such as green+yellow, etc).

In Table 4, we provide the Frechet Inception Distance (FID) of each method (the lower the better). We observe that our algorithm has the lowest FID score, i.e., our algorithm generates the most high-quality images from interventional distribution.

Table 4: Baseline comparison with FID scores

|  | Conditional | NCM | Our algorithm |
|---|---|---|---|
| FID Score | 67.012 | 71.646 | 43.507 |

Table 5: Color probability distribution of the sampled images

| Predicted color probabilities | Red | Green | Blue | Yellow | Magenta | Cyan |
|---|---|---|---|---|---|---|
| Conditional model: $P(Y\|X = red)$ | 0.1889 | 0.4448 | 0.1612 | 0.1021 | 0.0232 | 0.0798 |
| Ours: $P(Y\|do(X = red))$ | 0.1278 | 0.2288 | 0.2097 | 0.1445 | 0.1177 | 0.1715 |

ii) Correctness of the sampling distribution: We use a classifier to detect the color frequency from the generated images of the conditional model and our algorithm. We observe the probabilities in Table 5.

In the generated samples from the conditional model, the color of digit image $X$ and digit image $Y$ are correlated due to confounding through backdoor paths. For example, for a digit image $X$ with color as red, $Y$ takes a value from [Red, Green, Blue] with high probability. Thus, the conditional model does not have the ability to generate interventional samples. On the other hand, our algorithm generates samples from the interventional distribution $P(Y|do(X))$ and the generated samples chooses different colors:[Red, Green, Blue, Yellow, Magenta, Cyan] with almost the same probability.

Therefore, we show our performance with a close baseline NCM and illustrate our capability of sampling from high-dimensional interventional distribution.

### E.2    COVID X-RAY DATASET

Now we consider the Covid X-Ray Dataset. We first outline the steps we perform to preprocess our data and then discuss training details for each of the models we train.

### E.2.1    DATA PREPROCESSING

Note that a full pipeline of data preparation is contained in `cxray/prep_cxray_dataset.sh` in the provided codebase. We start by downloading the corpus approximately 30K Covid X-Ray images[3]. Then we download Covid-19 labels and Pneumonia labels[4] and attach labels to each image. Then we convert each image to black-white one-channel images and rescale each to a size of $(128 \times 128)$ pixels. Finally, a random split of the 30K images is performed: keeping 20K to be used during training, and 10K to be used as validation images. Note that the labels come with a set of 400 test images, but 400 images is too small to be an effective test set (say for FID computations). We will be explicit about where we use each data set.

### E.2.2    DIFFUSION TRAINING DETAILS

We train a diffusion model to approximate $P(X|C)$. To do this, we use a standard UNet and classifier-free guidance scheme. We train for 100K training steps over the 20K-sized training set, using a batch size of 16. This takes roughly 10 hours on a single A100. The same classifier-free guidance parameters are used as in the NapkinMNIST diffusion training.

### E.2.3    CALIBRATED CLASSIFIER TRAINING DETAILS

To train a classifier to sample from $P(N|C, X)$, we note that our inputs are a $(1, 128, 128)$ image and a binary variable. Our architecture is as follows: we create an embedding of $X$ by modifying the final linear layer of a ResNet18 to have output dimension 64 (vs 1000), and modify the input channels to be 1. We create an embedding for $N$ by using a standard embedding layer for binary variables, with embedding dimension 64. These embeddings are then concatenated and pushed through 3 fully

---
[3]https://www.kaggle.com/datasets/andyczhao/covidx-cxr2
[4]https://github.com/giocoal/CXR-ACGAN-chest-xray-generator-covid19-pneumonia

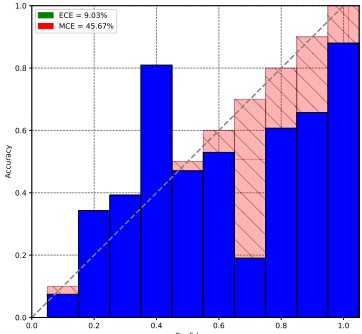 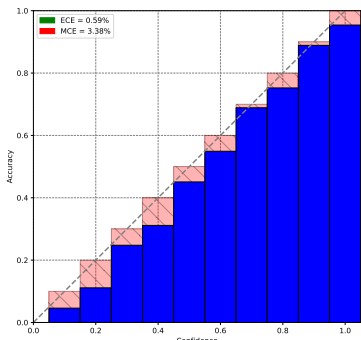

Figure 9: **Reliability plots for $\mathbf{P(N|C,X)}$:**. Reliability plots which overlay accuracy versus classifier confidence. (Left) Reliability plot for $P(N|C,X)$ without calibration. (Right) Reliability plot for $P(N|C,X)$ with temperature scaling calibration applied.

connected layers with ReLU nonlinearities and dimension 128. A final fully-connected layer with 2 outputs is used to generate logits.

Training follows a standard supervised learning setup using cross entropy loss. We train for 100 epochs using a batch size of 256 and a standard warmup to decaying learning rate (see code for full details).

We note the deep literature suggesting that even though classifiers seek to learn $P(N|X,C)$, neural networks trained using cross entropy loss do not actually do a very good job of estimating this distribution. Training attains an accuracy of $91.2\%$ on the test set. Calibrated classification seeks to solve this problem by modifying the network in a way such that it more accurately reflects this distribution. We follow the standard approach using temperature scaling of the logits, where a temperature is learned over the validation set, using LBFGS with a learning rate of 0.0001 and a maximum of 10K iterations. This does not affect the test accuracy at all, but drastically improves the ECE and MCE reliability metrics. See Guo et al. (2017) for a further discussion of temperature scaling.

