# OpenReview forum: "Conditional Generative Models are Sufficient to Sample from Any Causal Effect Estimand"
_ICLR.cc/2024/Conference — Submitted to ICLR 2024_

### Official Review · Reviewer_ZLmN · 2023-10-26

**Soundness:** 3 good
**Presentation:** 3 good
**Contribution:** 2 fair
**Rating:** 6
**Confidence:** 2

**Summary:**

This addresses the challenge of estimating causal effects, which typically requires access to conditional likelihoods. In high-dimensional scenarios, this can be problematic. The paper introduces a novel approach using conditional generative models to compute identifiable causal effects in graphs with latent confounders. The authors also present a diffusion-based method for sampling from interventional distributions. Experimental results are demonstrated on synthetic and real-world datasets, showing the algorithm's utility.

**Strengths:**

- The writing in this paper is quite clear, and the organization is well-structured. The theoretical foundation of the method is solid.
- The problem studied in the paper is quite interesting.

**Weaknesses:**

- While using a causal graph to model the relationships between statistical variables, deriving identifiability conditions is crucial. However, when extending this to high-dimensional data, the theory is elegant, but the practical applications are quite tricky. For instance, as seen in the experiments within the paper, they are entirely simulated cases. I find this to be not very practical.
- If the authors could provide some simulation data in the experimental section, it would better support the conclusions in the paper. This is because, when relying solely on image data, evaluation metrics can sometimes be less accurate.
- Could the authors provide some failure cases, where the situations that would occur if the assumptions in the paper are not met?

**Questions:**

I'm familiar with DAG learning and effect estimation, but I don't have a deep understanding of ADMG. Therefore, I find it challenging to provide an assessment of the method's innovation at this point. I will rely on the opinions of other reviewers to evaluate this aspect later on.

---

> ### Author Response · Authors · 2023-11-18
> **Response to Reviewer ZLmN (1/2)**
>
> We would like to thank the reviewer for their comments and feedback and are happy to learn that the reviewer found our
> paper well-structured and our method theoretically solid.
>
> ## Practical applications with high-dimensional data
>
> > When extending this to high-dimensional data, the theory is elegant, but the practical applications
> > are quite tricky. For instance, as seen in the experiments within the paper, they are entirely simulated cases. I find
> > this to be not very practical.
>
> We thank the reviewer for their comment. We humbly point the reviewer to section 5.2 of our paper where we shared our
> algorithm performance of
> an experiment on the real-world COVID
> X-RAY dataset (https://www.kaggle.com/datasets/andyczhao/covidx-cxr2/data). However, if
> the reviewer could kindly suggest any challenging applications or datasets suitable for causal inference,
> we would be very happy to evaluate our algorithm on those.
>
> For the reviewer's convenience, here we share our real-world experiment in detail.
> The COVID X-RAY contains 30k X-ray images and COVID and pneumonia labels.
> With this dataset, we associated the front door causal graph:
> $$\mathrm{Covid Symptoms} \rightarrow \mathrm{Xray Images} \rightarrow \mathrm{Pneumonia Detection};
> \mathrm{Covid Symptoms} \leftarrow U \rightarrow \mathrm{Pneumonia Detection}$$
> where $U$ is an unobserved variable (patient/hospital location).
> Our aim was to generate high-quality samples from $P(\mathrm{Xray images}| do(\mathrm{Covid Symptoms}))$ and estimate
> $P(\mathrm{Pneumonia Detection}|do(\mathrm{Covid Symptoms}))$, i.e., how likely a patient will be diagnosed with
> Pneumonia if we intervene on the COVID-19 symptoms, assessing the causal effect.
>
> For the reviewer's convenience, here we share part of the section 5.2 experiment results. Let N= Pneumonia detection and
> C = Covid Symptoms.
>
> ### Result 1:
>
> We evaluated the image quality of the diffusion model approximating $P(X|C)$. Below we provide the Frechet Inception
> Distance (FID)
> scores of generated image samples compared to the true image samples. Here, `Generated C=c`, means we sample from
> the diffusion model conditioned on the value $C=c$. `Real (C=c)` refers to the corresponding image samples consistent
> with $C=c$. Low values on the diagonal and high values on the off-diagonal imply we are sampling correctly from conditional
> distributions.
>
> Class-conditional FID scores for generated Covid XRAY images (lower is better).
>
> | FID scores of <br/>Generated vs Real image | Real: C = 0 | Real: C = 1 |
> |--------------------------------------------|-------------|-------------|
> | Generated: C = 0                           | 15.77       | 61.29       |
> | Generated C = 1                            | 101.76      | 23.34       |
>
> ### Result 2:
>
> We evaluate the query of interest $P_C(N)$. Since we do not have access to the ground truth of an interventional
> distribution
> for real data, we consider i) our evaluated
> $P_C(N)$ versus ii) an ablated version where we replace the diffusion sampling mechanism with $\hat P(X|C)$, where we
> randomly select an X-ray image from the validation set. We also consider the query $P_C(N)$ if there were iii) no
> latent confounders in the graph, in which case, the interventional query $P_C(N)$ is equal to $P(N|C)$.
>
> | $P_{C}(N=1)$     | c=0   | c=1   |
> |------------------|-------|-------|
> | i) Diffusion     | 0.622 | 0.834 |
> | ii) No Diffusion | 0.623 | 0.860 |
> | iii) No Latent   | 0.406 | 0.951 |
>
> ## Simulated experiments
>
> > If the authors could provide some simulation data in the experimental section, it would better support the conclusions
> > in the paper.
>
> We agree with the reviewer that it will be useful to include experiments with simulated data where variables are low
> dimensional discrete.
> Then we would be able to estimate $P_{X}(Y)$ and compare with the ground truth and evaluate them with different measures
> such as
> total variation distance (TVD) or KL divergence. Since our main target was to sample from the high-dimensional
> interventional
> distribution, we only considered image data during submission. Based on the reviewer's suggestion, we will add these new
> experiments with discrete data in the camera-ready version of our paper. If the reviewer has
> any specific suggestions about simulated datasets, we would be happy to include them.

---

> > ### Author Response · Authors · 2023-11-18
> > **Response to Reviewer ZLmN (2/2)**
> >
> > ### Failure cases
> >
> > > Could the authors provide some failure cases, where the situations that would occur if the assumptions in the paper
> > > are not met?
> >
> > Here we provide some failure cases that may occur if our assumptions are not satisfied:
> > 1. If the causal graph is incorrect and does not reflect the data-generating process properly, then we
> > might not get correct samples from the target interventional distribution, although please note that this is a common
> > assumption in
> > causality literature.
> > 2. If the conditional generative models are not correctly trained then the generated samples will not represent the
> > target interventional
> > distributions. Thus, we assume model convergence similar to many papers that apply deep-learning approaches.
> >
> > ### DAG vs ADMG
> >
> > For the reviewer's convenience, we provide a short description of ADMG here. In causal inference literature, ADMG refers
> > to acyclic directed mixed graphs. It is equivalent to a directed acyclic graph (DAG) where some variables
> > are unobserved (latent confounders cause two observed variables). For example, for our COVID experiment, $C \leftarrow
> > Xray \rightarrow N; C\leftarrow Patient Location \rightarrow N$
> > is a DAG. Now, suppose we can not observe patient location in the data, then we have an ADMG which we represent as
> > $C \leftarrow Xray \rightarrow N; C\leftrightarrow N$.
> >
> > We hope we addressed the reviewer's concerns. We are happy to answer any other questions the reviewer might have.

---

> > > ### Comment · Reviewer_ZLmN · 2023-11-22
> > >
> > > Thank you for providing detailed explanations. Based on the addressing of all my concerns, I have revised my rating to 6. I will also discuss the paper's contribution with other reviewers in the next discussion phase.

---

> > > > ### Author Response · Authors · 2023-11-22
> > > >
> > > > Dear Reviewer ZLmN,
> > > > We are happy to know that we have addressed your concerns. We cordially thank you for acknowledging our contributions and raising the score.
> > > >
> > > > -The authors.

---

### Official Review · Reviewer_2Jjc · 2023-10-31

**Soundness:** 3 good
**Presentation:** 2 fair
**Contribution:** 1 poor
**Rating:** 1
**Confidence:** 4

**Summary:**

The authors present an algorithm that allows for sampling from interventional distributions (using observational data & generative models). The algorithm is based on Shpitser & Pearl's revised ID algorithm with the modification that training data and conditional generative models (to be trained on said data) are discussed. The intuition behind said modification is being discussed using three examples. At the end, an empirical part on MNIST and semi-synthetic COVID/X-ray data are presented.

**Strengths:**

IMHO the paper's noteworthy strengths are limited to their idea rather than the execution, therefore, they should be considered (where applicable) as counterfactuals for the moment. Said "potential" strengths are considered one-by-one in the following list (the list is ordered in correspondence to the paper presentation):
* Precise coverage of necessary ideas within Pearl's causality framework for understanding the contribution (a.k.a. paper is good on the causality side of things)
* Use of examples with increasing difficulty to conceptualize the key idea from first principles. Usage of visual means (schematic illustrations).
* Discussion of Algorithm 1's steps
* Reasonable semi-synthetic extension for real-world data
* An effort of a self-enclosed treatise i.e., theory + empirics

**Weaknesses:**

TL;DR: Respectfully, the authors should not feel attacked by any of the following, mostly, I feel like ignoring many of the "weaknesses" but at the core of this section really stands the heavy overreliance on the causal side of things, even though the project is intended as a work on the intersection to ML. The lack of discussions on the other end essentially invalidates the key contribution, the ID-DAG algorithm, as simply being a copy of Shpitser & Pearl's simplified version of Tian's original algorithm, just that we have to use actual samples and models instead of magically having the actual probability distributions at hand. This is precisely what ML is in essence, and discussing characteristics w.r.t. learning, and not the causal part that we already know of by Shpitser, Pearl and others, would have been exactly this paper's key contribution.

The paper suffers from several disadvantages, ranging in importance from minor to more fundamental (and the minor ones, especially w.r.t. presentation, can be improved quickly). Thereby, the following list - again one-by-one - aims to provide specific pointers with improvement suggestions if applicable (please note, the list is unordered):

* While agreeing with the sentiment that estimating high dimensional, arbitrary conditional distributions to arbitrary precision is difficult, doing so in a more general sense is not and there exist different ways of handling this (just through Bayes' rule or through explicit modelling or through approximations). It comes to mind that in the introduction all Pearl related work (the causal side of things) are described precisely, whereas there is no reference whatsoever to works on the generative side of things apart from the handful of deep models. For example, following the difficulties encountered with Bayesian Networks's inference, advances were made on probabilistic circuits by Darwiche, Poon, Domingos, Perharz, Vergari, Van den Broeck and many others. In said realm, there are even first results on causality both w.r.t inference and sampling (as discussed in this work). To point the authors to concrete works, I link the following (sorted by date): NeurIPS 2021, "Interventional Sum-Product Networks: Causal Inference with Tractable Probabilistic Models" and AISTATS 2023 "Compositional Probabilistic and Causal Inference using Tractable Circuit Models". By now there are probably a lot more works out there, which is why I'd kindly encourage the authors to have an extensive look at all of these related works.
* The background section clearly suffers from overloading. Many concepts are needed for the work, however, this work should not be considered as an introductory lecture and given the space limitations IMHO they are the things (as opposed to others listed below that) should arguably be placed in the appendix in a more appropriate manner, covering only the highly relevant prerequisites in the background section.
* $X$ is not defined in Lemma 3.3.
* $P_x(y)$ is not defined when first introduced in Lemma 3.3.
* Please consider improving figure presentation alongside the following three dimensions: (i) proportions, especially Figures 2,3,4,5 suffer from for example small font sizes that make zooming a necessity, (ii) descriptions, legends for things like color codes within the figure & figure caption are missing but also generally the captions do not even work as capturing a "take-away message" let alone as being self-enclosed means of communicating the figure's ideas, and (iii) consistency, for example Figure 3 has \sum written out (as a not compiled normal text).
* Visual components, shaded quadrangle and circular node, are both not defined in Figure 3. Unlike the examples before, one cannot conclude anything (which is unfortunate since this is exactly the interesting part). Labels, as previously, for training and sampling phases are missing.
* As a fun nod, when looking closely since it is quite ironic, but DAGs are actually never defined. The paper begins with ADMGs and then abandons them through replacement with DAGs without any notice. Said DAGs further give a false sense of generality of the result since ADMGs are a generalization of DAGs.
* $\pi$ in Algorithm 1 is not defined.
* Key parts of the algorithm s.a. ConstructDAG or Step 7's modification are not being discussed in the main paper.
* Most severe contribution issue: the "key contribution" as the author's put it, Algorithm 1, is identical to Algorithm in Figure 3 of Ilya Shpitser and Judea Pearl "Identification of joint interventional distributions in recursive semi-Markovian causal models" (AAAI 2006) up to the sampling network part, which does not become apparent whatsoever through the paper's discussions. Furthermore, the lack of characterization nullifies this contribution and quickly renders the situation a pure application of these prior results but without an actual emphasis on the application part.
* Most severe writing issue: Theorem 4.1. is neither covered nor proven in the main paper. It is quite evident that the style of writing chosen by the authors (based on the insights from this whole section) is based on "exploiting" the appendix for more presentation space, this goes against the submission guidelines of ICLR (and while not as severe as something like non-anonymity, it could still warrant a discussion of desk rejection). I've thoroughly checked the appendix, however, it is important to note that this is not a requirement for the reviewer and specifically so because of the fact that otherwise the meaning of the 9-page main paper would be rendered meaningless. Personally, this does not bother me since I'm content-centric, however, this is a conflict with the conference's guidelines and needs to be covered.
* IDTrain in Section 5 is not defined.
* The experimental section evaluation is rather a protocol, than it is an interpretation/discussion of the observed results. The resolution on Fig.5 even on highest zoom does not allow a proper inspection. Still, it seems that "regular" diffusion-based issues arise as well (however, to be expected).


This final list is a list of suggestions with concrete ideas that can hopefully help the authors improve their work, as this review is intended as a means of constructive feedback i.e., to help the author's contribution be a great one for the community:

* Please consider using \citep{} for non-direct references like the ones in the introduction. Especially since the citations are not highlighted otherwise, rendering readability especially difficult.
* Please consider another pass (possibly by a non-author reader) to avoid writing mistakes, which are not easily detectable by common software. The manuscript various such instances, for example "international" instead of "interventional" in the Related Work section. Or also cases of punctuation such as for example a missing dot between the two sentences at beginning of Section 4.
* In the same way as for the regular text writing part, please careful check the mathematical notation. Similarly, there are also such instances where intentions don't comply with what is typeset, for example model $M$ instead of model $\mathcal{M}$ in Lemma 3.3.
* Avoid using notation in mathematical results if not needed, for example in Lemma 3.3. introducing $M$ is not necessary i.e., it is not being used by the result. For example rewrite as: "Let $G$ denote the causal graph entailed by some SCM."
* Please consider highlighting the two critical (and by the way rather strong) assumptions at the end Section 3.
* Please consider doing another pass also with respect to captions and such. For example in your "key contribution" the algorithm in Alg.1. the label for "causal graph" is missing at the Input part.


As a final remark on the score: if the ICLR reviewing scale where a 1-10 scale, then I'd opt for a score of 3. However, I only get to choose between 1 and 3 this time around and since the identified issues with this work are severe based on this review's assessment, the conclusion is a 1. To not end this on a negative note, though, IMHO the potential for this work is good and would be of good value for the community.

**Questions:**

TL;DR: No questions.

Even though several quantities throughout have not been defined, I'm confident in being able to guess what the authors actually mean, therefore, no questions are derived on that end. Furthermore, the lack of discussions on both the theoretical and empirical end regarding the non-causal side of things, that is so-defined sampling networks/generative models, does not allow me to raise any further questions.

---

> ### Author Response · Authors · 2023-11-18
> **Response to Reviewer 2Jjc (1/3)**
>
> We cordially thank the reviewer for their effort in writing a detailed review. Their feedback has greatly
> motivated us to improve our paper and more definitely establish its novelty compared to the current literature.
>
> We divide our responses into three main parts.
>
> ## Writing issues:
> ### Paper's presentation:
> We greatly appreciate the reviewer's effort in thoroughly checking our manuscript and pointing out the minor writing
> issues, and we deeply apologize for those. Based on the reviewer's suggestion, we made the following updates to our manuscript
> and will upload the updated version soon:
>
> * Definitions: We redefined $X$ and $P_{X}(Y)$ in Lemma 3.3 and $\pi$ in Algorithm 1. We would like to kindly point out
>   that our algorithm is for acyclic directed mixed graph (ADMG), we used the term DAG to refer to the set of trained
>   neural networks which have a different structure, edges, and even nodes compared to the original causal graph. To avoid
>   confusion, we replaced the "DAG"
>   term with a "network" of trained models.
> * Figures: We i) improved the objects in Figures 2,3,4 and 5 and mentioned the significance of those, and ii) increased the
>   readability of some texts. We iii) added more descriptive captions to assist the readers. We iv) fixed the uncompiled
>   equations in the figures.
> * Discussion: We added a new section (Appendix D) in our updated submission, where we described the ConstructDAG and the
>   step 7 of our algorithm in more detail.
> * Another pass: We made another pass through the paper and fixed the remaining minor issues mentioned by the reviewer.
>   Since ICLR policy allows to update the submitted paper with minor changes during rebuttal, we fixed all the writing
>   issues mentioned by the reviewer and will upload the new version as soon as possible. We will be very happy if the
>   reviewer does not finalize his decision about our paper based on these.
>
>
> ### Exploiting the appendix:
>
> > "Theorem 4.1 is neither covered nor proven in the main paper"
>
> We would like to humbly disagree with the reviewer's concern about us exploiting the appendix. The reviewer is right
> that the proofs are provided in Appendix B and C. However, to our knowledge, it is quite common in the ML community to
> postpone theorem proofs to the appendix. We also believe that the statement of Theorem 4.1 is self-explanatory as it
> only states soundness and completeness, again a well-understood notion in the community. As evidence that this practice
> does not violate ICLR guidelines, we refer to two ICLR-2023 accepted papers (top 5\%:
> openreview.net/pdf?id=7YfHla7IxBJ; openreview.net/pdf?id=l6CpxixmUg) where the authors not only put all their proofs in
> the appendix but also explained their theorems very little, if any. Therefore, we would kindly request the reviewer to
> reconsider his claim that we have "exploited" the appendix. We believe this unfortunate choice of words was unjustified.
> To consider the reviewer's feedback for improving our paper, we will add a proof sketch and elaborate on what soundness
> and completeness means in the main paper for Theorem 4.1.
>
>
> ## The role of the ID algorithm in our work:
>
> > The "key contribution" as the authors put it, Algorithm 1, is identical to Algorithm
> > In Figure 3 of Shpitser et al "Identification of joint interventional distributions in recursive
> > semi-Markovian causal models" (AAAI 2006) up to the sampling network part.
>
> We believe there is a severe misunderstanding, and we apologize if our writing was not clear enough.
> First, note that both the 2006 and 2008 papers below have the same algorithm (conference vs. journal versions):
>
> * Shpitser et al (2008) "Complete identification methods for the causal hierarchy."
> * Shpitser et al (2006) "Identification of joint interventional distributions in recursive semi-Markovian causal
>   models."
>
> But more importantly, our algorithm is not identical to Ilya and Pearl's algorithm.
> ID algorithm gives a way to step-by-step convert an interventional distribution to **some** function of the observational
> distribution.
> We ask ourselves, can we use this algorithm to convert any identifiable query to a sequence of forward-conditional
> networks.
> It is then natural that the sampling version should follow the steps of this algorithm and identify the necessary
> modifications to turn it into a sampling algorithm.
> Therefore, the ID algorithm is not a baseline but a crucial, necessary component of our solution. We cordially invite the
> reviewer to look at our contribution from this lens.
>
> For the high-level differences note that the ID algorithm is only practical for estimating the causal effect when variables are discrete. Whereas, we
> propose a sampling algorithm and adapt it to deal with high-dimensional variables using the ID algorithm as a guide.
> Throughout the paper, we attempted to precisely describe the limitations each step of the ID algorithm will face while
> dealing with high dimensional variables and how our proposed new techniques can resolve that.

---

> > ### Author Response · Authors · 2023-11-18
> > **Response to Reviewer 2Jjc (2/3)**
> >
> > ## The role of the ID algorithm in our work (cont.)
> > To be more precise, here we describe the major challenges the ID algorithm faces while dealing with high-dimensional
> > data and our contributions to solve them:
> >
> >
> > ### 1. Building the sampling network
> >
> > Step 4 of the ID algorithm splits the graph G(V), into multiple c-components $S_i, \forall_i$ and estimates $P_
> > {{v}\setminus s_i}(s_i)$ recursively. However, following this step directly and sampling from each c-component $P_
> > {{v}\setminus s_i}(s_i)$ in the topological (or any other)
> > order might not work since **it might create cycles in the sampling network**.
> >
> > For example, in $X \rightarrow W_1 \rightarrow W_2 \rightarrow Y; X \leftrightarrow W_2; W_1 \leftrightarrow Y$. Here,
> > we can not sample variables one-by-one following the order $\{X, W_1, W_2, Y\}$ since $\{X, W_2\}$ and $\{W_1, Y\}$ are
> > confounded and we have to sample them together, i.e., sample $\{X, W_2\}$ or sample $\{W_1, Y\}$. But which one to
> > sample first? If we consider sampling $\{X, W_2\}$ first, we need $W_1$ as input to the model to generate $W_2$ since
> > $W_1$ is $W_2$'s parent. If we consider sampling $\{W_1, Y\}$ first, we need both $X_1$ and $W_2$ as input to the model
> > for the same reason. This creates a cyclic issue.
> >
> > Our method solves this challenge by training corresponding models to each c-component and merging those trained models
> > to build a single sampling network. In other words, while traversing the recursive steps of the ID algorithm, we stay in
> > the DAG realm and only train (no sampling) the required models and carefully merge them to avoid cycles in the sampling network. After finishing all the steps, we use the
> > final merged sampling network to sample from the interventional distribution. Our approach makes sure that variable
> > values are consistent in each sample and respect the interventional distribution.
> >
> > ### 2. New interventional data generation:
> >
> > Step 7 of the ID algorithm updates the probability distribution $P$ to a new argument value for the next recursive call.
> > It is non-intuitive how to modify this step if we aim to generate interventional samples and not only estimate the
> > causal effect.
> >
> > We modify this step by generating a new interventional dataset using required conditional models that are trained on
> > observational data. This newly generated interventional dataset is passed as the dataset argument for the next recursive
> > step of the algorithm. This new dataset will be utilized to train conditional models in the deeper level of the
> > recursion.
> >
> > ### 3. New arguments to maintain sampling consistency
> >
> > At step 7, we send a different set of arguments compared to the ID algorithm for the next recursive call. Suppose we
> > have two c-components $S_1$ and $S_2$ and $V$ is a set of all variables. While estimating $P_{V\setminus S_i}(S_i)$, the
> > ID algorithm drops some of the intervened variables $V\setminus S_i$ from the causal graph after they are applied for
> > estimating the causal effect.
> >
> > Unlike the ID algorithm, we keep the intervened variables connected in
> > the causal graph and utilize them as input conditions while training all conditional models at the deeper level of the
> > recursion. More precisely, we modify the ID algorithm's argument as $X\cap S'
> > \rightarrow X$ and $G_{S'} \rightarrow G_{S', \overline{X_{Z}}}$, i.e., we keep some extra variables in the intervention
> > argument and the causal graph argument.
> >
> > This allows us to sample from multiple c-components consistently respecting the distribution which is highly
> > non-intuitive from simple observation of the original ID-algorithm. We can now train conditional models in both $S_1$
> > and $S_2$ such that after merging, they can generate consistent samples.
> >
> > In summary, we only follow the recursive route of the ID algorithm. Our modifications at each step to deal with high
> > dimensional sampling highlight our contribution to this paper. We will more clearly describe these distinctions in our paper.

---

> > > ### Author Response · Authors · 2023-11-18
> > > **Response to Reviewer 2Jjc (3/3)**
> > >
> > > ## Novelty compared to the mentioned literature:
> > >
> > > > There is no reference whatsoever to works on the generative side of things apart from the handful of deep models.
> > > > For example, following the difficulties encountered with Bayesian Networks's inference, advances were made on
> > > > probabilistic circuits
> > > > ... In said realm, there are even first results on causality both w.r.t inference and sampling.
> > >
> > >
> > > We agree with the reviewer that causal reasoning with sum-product networks is an important research direction that can
> > > deal with the tractability issues associated with Bayesian networks. However, there is one major difference between the
> > > goal of these works and the goal of our paper. We are interested in enabling high-dimensional causal inference,
> > > including graphs that contain multiple image nodes, using finite data. Thus, our objective is to develop a method that
> > > has low **sample-complexity**. On the other hand, probabilistic circuits literature is interested in reducing **computational
> > > complexity** of the (causal) inference problem. This important difference separates us from them, in the sense that it is crucial for us to show that we can sample from any interventional distribution using only feedforward models. The results in the said literature do not subsume our results. For example, in the paper "Interventional Sum-Product Networks: Causal Inference with Tractable Probabilistic Models", the authors say:
> > >
> > > "The causal effect $p(y|do(x))$ in the napkin graph: $$W \rightarrow Z \rightarrow X \rightarrow Y; X \leftrightarrow W
> > > \leftrightarrow Y $$ is identified as $$p(y| do(x)) = \frac{\sum_{w} p(y,x|w,z) p(w)}{\sum_{w} p(x|w,z) p(w)}$$ using
> > > the do-calculus where each of the r.h.s. components can be modeled by (i)SPN respectively."
> > >
> > > Thus, the proposal is to model each probabilistic term in the expression with (i)SPNs. But this does not give us a way
> > > how to connect those consistently to samples from the overall distribution. Note that, if sampling each of the variables
> > > is not done in a consistent way, there might be cyclic dependency during the sampling process (as we showed above). The
> > > necessity of consistent sampling was not clear until our work. We show that the ID algorithm can be taken as a guide,
> > > but it has to be modified otherwise there will be cyclic dependencies during the sampling process.
> > >
> > > Although the above work mentions that they can quantify $p(y|do(x))$ as proven in Proposition 1 of their paper, they do
> > > not suggest how they can do that with only observational data which is known as identifiability in causality literature.
> > > Proposition 1 and all other proofs require access to both observational **and interventional** data. On the other hand, our
> > > the algorithm only requires observational data for training the conditional model, thus offering causal identifiability capability.
> > > Our result does not contradict the probabilistic circuits literature, but if anything, enables it by showing that we now
> > > have an algorithm that only needs feed-forward models. Again, this result was not known before, although it seems to
> > > have been vaguely claimed in the above paper.
> > >
> > > We will also add a discussion about the 2nd paper mentioned by the reviewer: "Compositional Probabilistic and Causal Inference using Tractable Circuit Models". This paper proposes an interesting method that can estimate causal effects efficiently by utilizing tractable probabilistic modeling. However, it is unclear if this method can utilize generative models such as diffusion models (similar to ours) to sample from high-dimensional interventional distributions (such as images).
> > >
> > > We will add these discussions in the main paper as well as cite all the papers in the probabilistic circuits community on
> > > causal inference. Again, we greatly appreciate the reviewer for pointing out this line of work to us. This allowed us to
> > > more clearly express the contribution of our work in regard to the related works.
> > >
> > > We hope we addressed the reviewer's major concerns about the writing and the contribution of our paper. We hope
> > > to have addressed other minor concerns as well. We would request the reviewer to re-evaluate their judgment of our paper in light of the new arguments we made in this response. We are very interested in and looking forward to having more discussions with the reviewer.
> > > Thank you!

---

### Official Review · Reviewer_Xy4K · 2023-11-01

**Soundness:** 2 fair
**Presentation:** 2 fair
**Contribution:** 2 fair
**Rating:** 5
**Confidence:** 3

**Summary:**

This paper proposes an approach that leverages conditional generative models to sample from identifiable interventional distributions. The method is validated using diffusion models on synthetic image data and a real-world COVID-19 chest X-ray dataset.

**Strengths:**

- The paper claims that their approach can be applied to any identifiable interventional distribution. This suggests a wide range of potential applications in different domains where causal inference is critical.

- The application of the method to a real-world COVID-19 chest X-ray dataset showcases its practical relevance in addressing a critical public health concern, highlighting its strength.

**Weaknesses:**

- The idea of sampling according to the topological order is not new. Most of the causal effects estimators (explicitly or implicitly) utilise this technique to draw samples from the interventional distribution using conditional distributions. For example, [1][2][3]

1/ Louizos, C., Shalit, U., Mooij, J. M., Sontag, D., Zemel, R., & Welling, M. (2017). Causal effect inference with deep latent-variable models. Advances in neural information processing systems, 30.

2/ Zhang, W., Liu, L., & Li, J. (2021, May). Treatment effect estimation with disentangled latent factors. In Proceedings of the AAAI Conference on Artificial Intelligence (Vol. 35, No. 12, pp. 10923-10930).

3/ Vo, T. V., Bhattacharyya, A., Lee, Y., & Leong, T. Y. (2022). An Adaptive Kernel Approach to Federated Learning of Heterogeneous Causal Effects. Advances in Neural Information Processing Systems, 35, 24459-24473.

Could the author please highlight the technical novelties of the proposed method?

- The experiments lack comparisons with baseline methods.

**Questions:**

1/ Is it possible to compare with some baselines?

2/ What are technical novelties of the proposed method?

---

> ### Author Response · Authors · 2023-11-18
> **Response to Reviewer Xy4K (1/3)**
>
> We thank the reviewer for their feedback and questions and appreciate them acknowledging the utility of our results in
> different domains and our practical relevance.
> Below we address all of the reviewer's concerns.
>
> ## Sampling according to the topological order is not sufficient:
>
> > The idea of sampling according to the topological order is not new. Most of the causal effects estimators (explicitly
> > or implicitly)
> > Utilize this technique to draw samples from the interventional distribution using conditional distributions.
>
> The reviewer raised an interesting concern that the idea of sampling according to the topological order is not new and
> we agree with them. However, this approach is **only feasible when there exists no unobserved confounders in the causal
> graph**.
> For graphs with confounders, sampling according to topological order has failure cases.
> Consider this causal graph:
> $$X \rightarrow W_1 \rightarrow W_2 \rightarrow Y; X \leftrightarrow W_2; W_1 \leftrightarrow
> Y.$$
> Here, we can not sample variables one-by-one following the order {$X,W_1,W_2,Y$} since {$X,W_2$} and {$W_1,
> Y$} are confounded and we have to sample them together, i.e., sample {$X, W_2$} or sample {$W_1, Y$}. But which one
> to sample first? If we consider sampling {$X, W_2$} first, we need $W_1$ as input to the model to generate $W_2$ since
> $W_1$ is $W_2$'s parent. If we consider sampling {$W_1, Y$} first, we need both $X_1$ and $W_2$ as input to the model
> for the same reason. This creates a cycle in the sampling network.
>
> Our algorithm takes several steps to avoid this issue for any causal graph and identifiable interventional query. For
> example, we train models for {$X, W_2$} considering all possible values of $W_1$ as input. Similarly, we train models for the 2nd c-component considering
> all possible values of its input variables. Finally, we connect the trained models from two c-components to build a
> single sampling network.
> In this sampling network, we do not have to sample any variables together with another one, and we can follow the topological order to
> sample all variables.
> Therefore, we would state that sampling according to the topological ordering is one single part of our whole solution.
> There are more subtleties such as:
> *  Building a sampling network: we carefully follow the steps of the ID algorithm to recursively divide the problem into
> c-components, train corresponding models, and build a consistent sampling network with the trained models.
> *  New interventional data generation: We generate new interventional datasets by training conditional models on
> observational data and utilize them in later steps.
> * Maintaining sampling consistency: We avoid the cyclic dependency issue mentioned above by strategically updating the
> causal graph and the intervention set.
>
> We describe all these steps in more detail in the next paragraph.

---

> > ### Author Response · Authors · 2023-11-18
> > **Response to Reviewer Xy4K (2/3)**
> >
> > ## Technical novelties of our paper
> >
> > > What are the technical novelties of the proposed method?
> >
> > We appreciate the reviewer for allowing us to highlight the strengths of our paper. Our main contribution is to show, by
> > construction, that one can sample from any identifiable interventional distribution only using observational data and
> > feed-forward, i.e., conditional generative models. Note that this was not known before our paper. Our construction uses
> > the ID algorithm to propose a sampling algorithm so that the new algorithm can deal with high-dimensional variables.
> > Here we describe the major technical differences of our algorithm:
> >
> > 1. **Building the sampling network**: At step 4 of our algorithm we split the variable set into multiple c-components and obtain corresponding models for each c-component recursively. We merge these trained models to build
> >    a single sampling network. While traversing the recursive steps of our algorithm, we stay in the DAG realm, i.e., no
> >    cycles in the sampling network, train the required models, and merge them. After finishing all the recursive steps, we sample using the final merged sampling network, which we show to be acyclic. This makes sure that variable values are consistent in each sample and respect the interventional distribution.
> >
> > 2. **New interventional data generation**: Step 7 of our algorithm requires us to generate a new interventional
> >    dataset from specific models that are trained on observational data. This newly generated interventional dataset is
> >    passed as the dataset argument for the next recursive step of the algorithm. This new dataset will be utilized to
> >    train conditional models corresponding to the deeper level of the recursion.
> >
> > 3. **New arguments to maintain sampling consistency**:
> >    Suppose we have two c-components $S_1$ and $S_2$ and $V$ is a set of all variables.
> >    While estimating $P_{V\setminus S_i}(S_i)$, the ID algorithm drops some of the intervened variables $V\setminus S_i$
> >    from the causal graph after they are applied to estimate the causal effect. Unlike the ID algorithm, we keep the
> >    intervened variables connected in
> >    the causal graph and utilize them as input conditions while training all conditional models at the deeper level of the
> >    recursion. This allows for sampling from multiple c-components in a consistent manner respecting the
> >    distribution. For example, models in both $S_1$ and $S_2$ will train considering each other's values such that during
> >    the sampling step, they can generate consistent samples.
> >
> > We precisely described these contributions in Appendix D in the updated submission.
> >
> > ## Difference from the related work reviewer mentioned
> >
> > [1], [2] and [3] propose interesting and novel approaches to solve the causal effect estimation
> > problem using variational autoencoders. However, their proposed solution and theoretical guarantees for causal
> > estimation are tailored for specific causal graphs containing treatment, effect, and covariates (or observed proxy
> > variables) where they apply the backdoor adjustment formula.
> >
> > On the other hand, we consider any arbitrary causal graphs (such as the popular napkin graph in Figure 3: $W_1 \rightarrow W_2 \rightarrow X \rightarrow Y;
> > X \leftrightarrow W_1 \leftrightarrow Y $)
> >  possibly containing confounders that require a more complicated identifiability expression (such as fractions: $p(y| do(x)) = \frac{\sum_{w_1} p(y,x|w_1,w_2) p(w_1)}{\sum_{w_1} p(x|w_1,w_2) p(w_1)}$) compared to the
> > adjustment formula. Our proposed solution can sample from any identifiable query by training diffusion models regardless
> > of how complicated the identifiability expression is.
> >
> > [1] Louizos, C., Shalit, U., Mooij, J. M., Sontag, D., Zemel, R., & Welling, M. (2017). Causal effect inference with
> > deep
> > latent-variable models. Advances in neural information processing systems, 30.\
> > [2] Zhang, W., Liu, L., & Li, J. (2021, May). Treatment effect estimation with disentangled latent factors. In
> > Proceedings of the AAAI Conference on Artificial Intelligence (Vol. 35, No. 12, pp. 10923-10930).\
> > [3] Vo, T. V., Bhattacharyya, A., Lee, Y., & Leong, T. Y. (2022). An Adaptive Kernel Approach to Federated Learning of
> > Heterogeneous Causal Effects. Advances in Neural Information Processing Systems, 35, 24459-24473.

---

> > > ### Author Response · Authors · 2023-11-18
> > > **Response to Reviewer Xy4K (3/3)**
> > >
> > > ## New experiment with Baseline comparison
> > >
> > > > Is it possible to compare with some baselines?
> > >
> > > According to the reviewer's suggestion, we compared our performance with two baselines and now provide the results below:
> > > We updated our submission and provided the details in Appendix E.1.5.
> > >
> > > For the reviewer's convenience, we provide a short description of the ground truth first.
> > >
> > > ### Ground truth:
> > >
> > > The causal graph:
> > > $$W_1 \rightarrow W_2 \rightarrow X \rightarrow Y;
> > > X \leftrightarrow W_1 \leftrightarrow Y $$
> > > Dataset:
> > > $W_1, X $ and $Y$ are digit images and $W_2$ are discrete properties (thickness and color) of $W_1$.
> > > Each digit image can take a value from [0,9] and a color from [Red, Green, Blue, Yellow, Magenta, Cyan].
> > >
> > > ### Baselines:
> > >
> > > We compared our algorithm with two baselines:
> > > Baseline 1: A classifier-free diffusion model that samples from the conditional distribution: $Y\sim P(Y|X)$.
> > > Baseline 2: The NCM algorithm (Xia et al 2021) that samples from the interventional distribution $P(Y|do(X))$.
> > >
> > > ### Algorithm execution:
> > >
> > > To easily convey the distinction between different methods, we chose two images i) of digit 3 and ii) of digit 5, both
> > > colored red
> > > and use them as the intervention value for $X$. Then we used these images to sample from the corresponding distributions $P(Y|X)$
> > > and $P(Y|do(X))$ using different methods. Although the implementation of these methods is different,
> > > we considered the performance of each method after running them for 300 epochs. The NCM algorithm took around 50 hours
> > > to complete while our algorithm took approximately 16 hours to complete.
> > >
> > > ### Results:
> > >
> > > We would kindly request the reviewer to look at Figure 8 in Appendix E.1.5 for the results. For the reviewer's
> > > convenience,
> > > we describe the results in detail here as well.
> > >
> > > **Image quality**: First, we discuss the image quality of the generated samples from each approach.
> > > The conditional model and our algorithm both generate high-quality images of digit 3 and digit 5 with a specific color from
> > > the six possible colors. While the colors in generated images from the NCM algorithm can not remain at any specific
> > > value and get blended (such as green+yellow etc).
> > >
> > > Below we provide the Frechet Inception Distance (FID) of each method (the lower the better).
> > >
> > > |           | Conditional | NCM    | Our algorithm |
> > > |-----------|-------------|--------|---------------|
> > > | FID Score | 67.012      | 71.646 | 43.507        |
> > >
> > > We observe that our algorithm has the lowest FID score, i.e., our algorithm generates the most high-quality images from
> > > interventional distribution.
> > >
> > > **Correctness of the sampling distribution**
> > > We use a classifier to detect the color frequency from the generated images of the conditional model and our algorithm.
> > > We observe the following probabilities.
> > >
> > > | Predicted  color probabilities       | Red    | Green  | Blue   | Yellow | Magenta | Cyan   |
> > > |--------------------------------------|--------|--------|--------|--------|---------|--------|
> > > | Conditional model: P(Y &#124; X=red) | 0.1889 | 0.4448 | 0.1612 | 0.1021 | 0.0232  | 0.0798 |
> > > | Ours: P(Y &#124; do(X=red))          | 0.1278 | 0.2288 | 0.2097 | 0.1445 | 0.1177  | 0.1715 |
> > >
> > > In the generated samples from the conditional model,
> > > the color of digit image $X$ and digit image $Y$ are correlated due to confounding through backdoor paths.
> > > For example, for a digit image $X$ with color as red, $Y$
> > > takes a value from [Red, Green, Blue] with high probability. Thus, the conditional model does not have the ability to
> > > generate interventional samples.
> > > On the other hand, our algorithm generates samples from the interventional distribution $P(Y|do(X))$ and the generated
> > > samples chooses different colors:[Red, Green, Blue, Yellow, Magenta, Cyan] with almost the same probability, which is
> > > the expected result given the ground-truth SCM.
> > >
> > > Therefore, we show our performance with a close baseline NCM and illustrate our capability of sampling from
> > > high-dimensional
> > > interventional distribution.
> > >
> > > [Xia et al 2021.]  Kevin Xia, Kai-Zhan Lee, Yoshua Bengio, and Elias Bareinboim. The causal-neural connection:
> > > Expressiveness, learnability, and inference. Advances in Neural Information Processing Systems, 34:10823–10836,
> > >
> > > We hope to have addressed all of the reviewer's concerns. We are looking forward to having more discussions with the
> > > reviewer. If the reviewer has further experiment suggestions and baselines we would be happy to add them to
> > > camera-ready.

---

### Official Review · Reviewer_jtV4 · 2023-11-07

**Soundness:** 3 good
**Presentation:** 2 fair
**Contribution:** 2 fair
**Rating:** 5
**Confidence:** 3

**Summary:**

The paper proposes ID-DAG as an algorithm to turn any causal estimand into a set of conditional generative models that can be used to estimate the causal query. The paper tests the algorithm on a synthetic MNIST task as well as a covid chest x-ray dataset and shows promising results.

**Strengths:**

The paper tackles the important problem of causal estimation in the presence of high-dimensional variables. It proposes a sound algorithm to efficiently estimate the causal query by expanding upon the ID algorithm. The experiments tackle interesting causal settings that would be difficult to solve with current causal estimation methods and show reasonable results.

**Weaknesses:**

The paper's presentation could be a little clearer with more precise definitions of the notation (e.g. $p(...\mid do(x=..))$ vs $p_x(...)$). Even though the paper cites [1] it simply states that they do not handle high-dimensional variables. It would be interesting to see a comparison of both methods to get a better understanding of the performance of ID-DAG. Generally, the evaluation is fairly limited and could be expanded upon. It mentions that the groundtruth interventional distribution in the case of MNIST is not accessible even though the data is synthetically generated. I recommend looking at a evaluation with MorphoMNIST [2] (and cite the package).
Additionally, it would be nice to spell out the differences of this algorithm in case of no unobserved confounding.

[1] Kevin Xia, Kai-Zhan Lee, Yoshua Bengio, and Elias Bareinboim. The causal-neural connection: Expressiveness, learnability, and inference. Advances in Neural Information Processing Systems, 34:10823–10836, 2021.
[2] Castro, Daniel C., et al. "Morpho-MNIST: quantitative assessment and diagnostics for representation learning." Journal of Machine Learning Research 20.178 (2019): 1-29.

**Questions:**

- It seems that most equations are written assuming discrete variables. Why is that?
- Fig. 2 uses $Y_2$ while the text mentions $Y$ - is this a typo?
- What does it mean for a digit to be thin or thick?
- For the covid graph, in the real world wouldn't we assume an edge from C->N as well?
- The evaluation generally could be more thorough. Things that could support this would be observational / interventional likelihoods or the evaluation that e.g. p(thickness) shouldn't change with a colour intervention.
- Please have a look at the usage of \citep vs \citet.

---

> ### Author Response · Authors · 2023-11-18
> **Response to Reviewer jtV4 (1/3)**
>
> We thank the reviewer for their feedback and are happy to hear that they found the problem we are solving interesting
> and the method we are proposing sound and efficient. Below we address all their concerns.
>
> ## Notation
>
> > The paper's presentation could be a little clearer with more precise definitions of the notation (e.g. $P(...|do(
> > x=...))$ vs $P_x(...)$}:
>
> We thank the reviewer for pointing it out. Both notations are meant to represent the same notion. We will make it more
> clear in our background section.
>
> ## MorphoMNIST package:
>
> > I recommend looking at an evaluation with MorphoMNIST [2] (and cite the package)
>
> We thank the reviewer for suggesting the MorphoMNIST [2] package
> and would like to happily share that we are already using this package from
> https://github.com/dccastro/Morpho-MNIST/ for data generation of our MNIST image experiment.
> In the napkin graph, three out of four variables are images: $W_1, X$, and $Y$. To generate these images we picked a random digit from the original MNIST dataset and thickened or thinned it using the MorphoMNIST package.
>
> Nonetheless, evaluating our algorithm performance with this package would be certainly an interesting step.
> For example: after training our models on the colored MNIST dataset, it would be interesting to observe how
> our models perform if we intervene with a digit image perturbed with swelling or fracture. How does using
> a set of diverse and unseen interventions affect the interventional samples generated from the algorithm? We would
> include these evaluations in the camera-ready version of the paper.
> We mentioned the URL in our paper, we will cite it now.
>
> ## Evaluation with the ground truth and
>
> > It mentions that the ground truth interventional distribution in the case of MNIST is not accessible even though the
> > Data is synthetically generated.
>
> This ground truth image dataset $D$ we generated corresponds to the observational distribution $P(W_1, X, Y)$.
> Thus, for specific image $X$ we know how image $Y\sim P(Y|X)$ will look like since we
> can check that in the observational dataset $D$. However, the ground truth dataset $D$ does not let us know,
> how image $Y\sim P(Y|do(X))$ would look like for a specific image $X$ since those are interventional samples.
> This is what we meant when we said that we do not have access to the ground truth interventional image data.
> However, we still evaluate our interventional samples. Please check below.
>
> Since we generated the images from some discrete properties such as digit$\sim[0,9]$, thickness$\sim[0,1]$, and color$\sim[0,5]$ (
> using MorphMNIST, etc), we have access to these discrete values. Therefore, we can use the original ID algorithm
> to estimate true $P(Y.color|do(X.color))$.
>
> After generating interventional image samples $Y\sim P(Y|do(X))$ with our models, we use a classifier to obtain the color of
> $X$ and the generated $Y$, i.e, $X.color$ and $Y.color$. Thus, we can estimate $P'(Y.color|do(X.color))$ from our
> generated interventional samples and compare against the true $P(Y.color|do(X.color))$. We showed this comparison in Table 1 in our paper.
>
> [2] Castro, Daniel C., et al. "Morpho-MNIST: quantitative assessment and diagnostics for representation learning."
> Journal of Machine Learning Research 20.178 (2019): 1-29.
>
> ## Our algorithm when no unobserved confounding
>
> > It would be nice to spell out the differences of this algorithm in case of no unobserved confounding.
>
> When there exists no unobserved confounder in the causal graph, each variable is an individual c-component of size 1.
> Thus, our algorithm will first visit step 4 and then directly go
> to step 6: the base case. At step 6, due to no confounding, we simply train one conditional model for each variable
> $V_i$ to match $P(V_i|V_{\pi}^{(i-1)}))$, i.e., the probability of $V_i$ given all variables appear before it in the topological
> order. We will make it more clear in the paper.
>
> When there exist confounders in the causal graph, we would have larger c-components, and we will recursively solve the
> problem for each c-component.
>
> ## Equations with discrete variables
>
> > It seems that most equations are written assuming discrete variables. Why is that?
>
> We followed the notation of the ID algorithm (Shpitser et al 2008), which is practical when variables are discrete,
> which is a major drawback. We used those equations to indicate which steps of
> the ID algorithm we are following and how we are replacing them. Our algorithm is not limited to discrete variables and can adapt to
> continuous high-dimensional probability distributions. We will clarify that the p(.) could refer to any probability measure, continuous or discrete.
>
> * Shpitser et al (2008) "Complete identification methods for the causal hierarchy."
>
> ## Fig. 2 typo:
>
> > Fig. 2 uses $Y_2$ while the text mentions $Y$ - is this a typo?
>
> We thank the reviewer for pointing out the typo. We apologize for this and already fixed it.

---

> > ### Author Response · Authors · 2023-11-18
> > **Response to Reviewer jtV4 (2/3)**
> >
> > ## Thin or thick image
> >
> > > What does it mean for a digit to be thin or thick?
> >
> > By thin or thick, we referred to the thickness of the image of the digit. A thick
> > digit image implies that the lines in that image will be comparatively thicker. Thin refers to the opposite. Please
> > check Figure 4.
> > The MorphoMNIST package (the one the reviewer suggested) allows one to perturb an image according to some properties
> > such as
> > thickness (high, low).
> >
> > ## $Covid \rightarrow Pneumonia$ in the COVID graph
> >
> > > For the covid graph, in the real world wouldn't we assume an edge from C->N as well?
> >
> > We apologize for being imprecise.
> > For the reviewer's convenience, we re-state the graph here: $C \rightarrow X \rightarrow N; C \leftrightarrow N$.
> > We considered C as the COVID symptoms and N as the pneumonia diagnosis by a healthcare specialist. We assumed that
> > pneumonia
> > diagnosis is feasible
> > through the inspection of an X-ray image ($X \rightarrow N$) and an Xray image is affected by the presence of Covid
> > symptoms ($C\rightarrow X$).
> >
> > Since COVID symptoms affect pneumonia diagnosis only through the inspection of the X-ray image by some specialists,
> > we assume no direct edge from COVID-19 to Pneumonia detection.
> > We will make it precise in our paper.
> >
> > ## Observational vs interventional evaluation
> >
> > We thank the reviewer for this suggestion. The color and thickness are unobserved variables in the causal graph.
> > Therefore, we can not intervene on them. However, we intervened on $X$ (digit image) observed the effect on $Y$, and
> > compared the corresponding images with images sampled from $P(Y|X)$. We added a figure (Figure 8) in Appendix E.1.5.
> > and illustrated the distinction between observational samples and interventional
> > samples. Below, we describe it in more detail.

---

> > > ### Author Response · Authors · 2023-11-18
> > > **Response to Reviewer jtV4 (3/3)**
> > >
> > > ## Comparison with Xia et al 2021:
> > >
> > > > It would be interesting to see a comparison of both methods to get a better understanding of the performance of
> > > > ID-DAG.
> > >
> > > We appreciate the reviewer for suggesting NCM (Xia et al 2021) as a baseline.
> > > According to the reviewer's suggestion, we compared our performance with NCM and another baseline and provided the
> > > results
> > > below:
> > > We updated our submission and provided the details in Appendix E.1.5.
> > >
> > > For the reviewer's convenience, we provide a short description of the ground truth first.
> > >
> > > ### Ground truth:
> > >
> > > The causal graph:
> > > $$W_1 \rightarrow W_2 \rightarrow X \rightarrow Y;
> > > X \leftrightarrow W_1 \leftrightarrow Y $$
> > > Dataset:
> > > $W_1, X $ and $Y$ are digit images and $W_2$ are discrete properties (thickness and color) of $W_1$.
> > > Each digit image can take a value from [0,9] and color from [Red, Green, Blue, Yellow, Magenta, Cyan].
> > >
> > > ### Baselines
> > >
> > > We compared our algorithm with two baselines:
> > > Baseline 1: A classifier-free diffusion model that samples from the conditional distribution: $Y\sim P(Y|X)$.
> > > Baseline 2: The NCM algorithm (Xia et al 2021) that samples from the interventional distribution $P(Y|do(X))$.
> > >
> > > ### Algorithm execution
> > >
> > > To easily understand the distinction among different methods, we chose two images i) of digit 3 and ii) of digit 5, both
> > > colored red
> > > as the intervention value for $X$. Then we used these images to sample from the corresponding distribution $P(Y|X)$
> > > and $P(Y|do(X))$ distributions using different methods. Although the implementation of these methods is different,
> > > we considered the performance of each method after running them for 300 epochs. The NCM algorithm took around 50 hours
> > > to complete while our algorithm took approximately 16 hours to complete.
> > >
> > > ### Results
> > >
> > > We would request the reviewer to look at Figure 8 in Appendix E.1.5 to get visual observation. For the reviewer's
> > > convenience,
> > > we describe the results in detail here as well.
> > >
> > > **Image quality**: First, we discuss the image quality of the generated samples from each approach.
> > > The conditional model and our algorithm both generate high-quality images of digit 3 and digit 5 with a specific color
> > > from
> > > the six possible colors. While the colors in generated images from the NCM algorithm can not remain at any specific
> > > value and
> > > get blended (such as green+yellow etc).
> > >
> > > Below we provide the Frechet Inception Distance (FID) of each method (the lower the better).
> > >
> > > |           | Conditional | NCM    | Our algorithm |
> > > |-----------|-------------|--------|---------------|
> > > | FID Score | 67.012      | 71.646 | 43.507        |
> > >
> > > We observe that our algorithm has the lowest FID score, i.e., our algorithm generates the most high-quality images from
> > > interventional distribution.
> > >
> > > **Correctness of the sampling distribution**
> > > We use a classifier to detect the color frequency from the generated images of the conditional model and our algorithm.
> > > We observe the following probabilities.
> > >
> > > | Predicted  color probabilities       | Red    | Green  | Blue   | Yellow | Magenta | Cyan   |
> > > |--------------------------------------|--------|--------|--------|--------|---------|--------|
> > > | Conditional model: P(Y &#124; X=red) | 0.1889 | 0.4448 | 0.1612 | 0.1021 | 0.0232  | 0.0798 |
> > > | Ours: P(Y &#124; do(X=red))          | 0.1278 | 0.2288 | 0.2097 | 0.1445 | 0.1177  | 0.1715 |
> > >
> > > In the generated samples from the conditional model,
> > > the color of digit image $X$ and digit image $Y$ are correlated due to confounding through backdoor paths.
> > > For example, for a digit image $X$ with color as red, $Y$
> > > takes a value from [Red, Green, Blue] with high probability. Thus, the conditional model does not have the ability to
> > > generate interventional samples.
> > > On the other hand, our algorithm generates samples from the interventional distribution $P(Y|do(X))$ and the generated
> > > samples chooses different colors:[Red, Green, Blue, Yellow, Magenta, Cyan] with almost the same probability.
> > >
> > > Therefore, we show our performance with a close baseline NCM and illustrate our capability of sampling from
> > > high-dimensional
> > > interventional distribution.
> > >
> > > [Xia et al 2021.]  Kevin Xia, Kai-Zhan Lee, Yoshua Bengio, and Elias Bareinboim. The causal-neural connection:
> > > Expressiveness, learnability, and inference. Advances in Neural Information Processing Systems, 34:10823–10836,
> > >
> > >
> > > We hope we addressed all of the reviewer's concerns. We are happily looking forward to having more discussions with them.

---

### Author Response · Authors · 2023-11-21
**Rebuttal phase discussion**

Dear AC and reviewers,

Thank you once again for the time and effort you spent on our paper so far. We greatly appreciate the feedback from all the reviewers. We have posted our replies which:

1. better delineate our contribution in relation to the existing work in probabilistic circuits, the ID algorithm, and other works on interventional sample generation,

2. demonstrate the advantage of our algorithm with new baseline experiments, and

3. clarify minor points that led to misunderstandings of our contribution.

We would like to kindly request the reviewers to take a look at our detailed responses. We hope that we addressed the reviewer’s questions and that they would reconsider their evaluation and score of our paper.

-The authors

---

### Meta-Review · Area_Chair_iSr6 · 2023-12-14

**Metareview:**

The paper develops a technique for using neural models (conditional generative models) to sample from different causal estimands given a causal graph. The reviewers are mostly lukewarm learning towards negative. The high level idea is interesting to transform causal estimands into things that just require forward sampling rather than some arbitrary property of the observational distribution. However, the position and clarity with respect to the related work needs to be improved. The most familiar reviewer was the most strongly against the paper likely because of clarity issues.

**Justification For Why Not Higher Score:**

The clarity issues on how the approach makes use of the existing ID algorithm.

**Justification For Why Not Lower Score:**

N/A

---

### Decision · Program_Chairs · 2024-01-16

Reject